# Mitochondrial CoQ deficiency is a common driver of mitochondrial oxidants and insulin resistance

Daniel J Fazakerley[1], Rima Chaudhuri[1], Pengyi Yang[2], Ghassan J Maghzal[3], Kristen C Thomas[1], James R Krycer[1], Sean J Humphrey[1], Benjamin L Parker[1], Kelsey H Fisher-Wellman[4], Christopher C Meoli[1], Nolan J Hoffman[1‡], Ciana Diskin[1], James G Burchfield[1], Mark J Cowley[5], Warren Kaplan[6], Zora Modrusan[7], Ganesh Kolumam[7§], Jean YH Yang[2], Daniel L Chen[8], Dorit Samocha-Bonet[8], Jerry R Greenfield[8], Kyle L Hoehn[9], Roland Stocker[3,10†*], David E James[1,11†*]

[1]Charles Perkins Centre, School of Life and Environmental Sciences, University of Sydney, Camperdown, Australia; [2]School of Mathematics and Statistics, University of Sydney, Camperdown, Australia; [3]Vascular Biology Division, Victor Chang Cardiac Research Institute, Darlinghurst, Australia; [4]Duke Molecular Physiology Institute, Duke University School of Medicine, Durham, United States; [5]The Kinghorn Cancer Centre, Garvan Institute of Medical Research, Darlinghurst, Australia; [6]Peter Wills Bioinformatics Centre, Garvan Institute of Medical Research, Darlinghurst, Australia; [7]Genentech Inc., South San Francisco, United States; [8]Garvan Institute of Medical Research, Darlinghurst, Australia; [9]School of Biotechnology and Biomedical Sciences, University of New South Wales, Sydney, Australia; [10]St Vincent's Clinical School, University of New South Wales, Sydney, Australia; [11]Charles Perkins Centre, Sydney Medical School, University of Sydney, Camperdown NSW, Australia

*For correspondence:
R.Stocker@victorchang.edu.au
(RS);
david.james@sydney.edu.au (DEJ)

†These authors contributed equally to this work

Present address: ‡Centre for Exercise and Nutrition, Mary MacKillop Institute for Health Research, Australian Catholic University, Melbourne, Australia; §Calico Labs, South San Francisco, United States

**Abstract** Insulin resistance in muscle, adipocytes and liver is a gateway to a number of metabolic diseases. Here, we show a selective deficiency in mitochondrial coenzyme Q (CoQ) in insulin-resistant adipose and muscle tissue. This defect was observed in a range of in vitro insulin resistance models and adipose tissue from insulin-resistant humans and was concomitant with lower expression of mevalonate/CoQ biosynthesis pathway proteins in most models. Pharmacologic or genetic manipulations that decreased mitochondrial CoQ triggered mitochondrial oxidants and insulin resistance while CoQ supplementation in either insulin-resistant cell models or mice restored normal insulin sensitivity. Specifically, lowering of mitochondrial CoQ caused insulin resistance in adipocytes as a result of increased superoxide/hydrogen peroxide production via complex II. These data suggest that mitochondrial CoQ is a proximal driver of mitochondrial oxidants and insulin resistance, and that mechanisms that restore mitochondrial CoQ may be effective therapeutic targets for treating insulin resistance.
DOI: https://doi.org/10.7554/eLife.32111.001

## Introduction

Insulin resistance is a major risk factor for several metabolic diseases including type two diabetes. This defect is found in all metabolic tissues most notably adipose tissue, muscle and liver. While insulin resistance in adipose and muscle tissue is likely to occur in a cell autonomous manner recent

**eLife digest** After we eat, our blood sugar levels increase. To counteract this, the pancreas releases a hormone called insulin. Part of insulin's effect is to promote the uptake of sugar from the blood into muscle and fat tissue for storage. Under certain conditions, such as obesity, this process can become defective, leading to a condition known as insulin resistance. This condition makes a number of human diseases more likely to develop, including type 2 diabetes. Working out how insulin resistance develops could therefore unveil new treatment strategies for these diseases.

Mitochondria – structures that produce most of a cell's energy supply – appear to play a role in the development of insulin resistance. Mitochondria convert nutrients such as fats and sugars into molecules called ATP that fuel the many processes required for life. However, ATP production can also generate potentially harmful intermediates often referred to as 'reactive oxygen species' or 'oxidants'. Previous studies have suggested that an increase in the amount of oxidants produced in mitochondria can cause insulin resistance.

Fazakerley et al. therefore set out to identify the reason for increased oxidants in mitochondria, and did so by analysing the levels of proteins and oxidants found in cells grown in the laboratory, and mouse and human tissue samples. This led them to find that concentrations of a molecule called coenzyme Q (CoQ), an essential component of mitochondria that helps to produce ATP, were lower in mitochondria from insulin-resistant fat and muscle tissue. Further experiments suggested a link between the lower levels of CoQ and the higher levels of oxidants in the mitochondria. Replenishing the mitochondria of the lab-grown cells and insulin-resistant mice with CoQ restored 'normal' oxidant levels and prevented the development of insulin resistance.

Strategies that aim to increase mitochondria CoQ levels may therefore prevent or reverse insulin resistance. Although CoQ supplements are readily available, swallowing CoQ does not efficiently deliver CoQ to mitochondria in humans, so alternative treatment methods must be found. It is also of interest that statins, common drugs taken by millions of people around the world to lower cholesterol, also lower CoQ and have been reported to increase the risk of developing type 2 diabetes. Further research is therefore needed to investigate whether CoQ might provide the link between statins and type 2 diabetes.

DOI: https://doi.org/10.7554/eLife.32111.002

evidence suggests that liver insulin resistance may occur via a mechanism involving defects in adipose tissue (*Perry et al., 2015*; *Titchenell et al., 2016*).

A range of perturbations have been shown to trigger insulin resistance including diets high in fat and/or sucrose (*Boden et al., 2015*; *Samocha-Bonet et al., 2012*; *Turner et al., 2013*), hyperinsulinaemia, hyperlipidaemia (*Roden et al., 1996*), inflammation (*Hotamisligil et al., 1994*), corticosteroids (*Houstis et al., 2006*; *Kusunoki et al., 1995*). These insults may cause insulin resistance via distinct means indicating that insulin resistance maybe a heterogeneous disorder. For example, stresses such as lipotoxicity (*Chavez et al., 2003*; *Griffin et al., 1999*), endoplasmic reticulum stress (*Ozcan et al., 2004*), mitochondrial dysfunction (*Kelley et al., 2002*; *Montgomery and Turner, 2015*) and mitochondrial oxidative stress (*Anderson et al., 2009*; *Hoehn et al., 2009*; *Houstis et al., 2006*) have all been reported to play a causal role in insulin resistance. However, we and others have shown that mitochondrial oxidative stress is a common feature of many in vitro insulin resistance models (*Hoehn et al., 2008*; *Houstis et al., 2006*) and metabolic tissues both from mice and humans (*Anderson et al., 2009*; *Paglialunga et al., 2015*), and in humans (*Anderson et al., 2009*). Further, many cellular stresses associated with insulin resistance such as ceramides (*García-Ruiz et al., 1997*) and endoplasmic reticulum stress (*Malhotra and Kaufman, 2007*) may also increase mitochondrial oxidant production. Despite this evidence for mitochondrial oxidants being a common feature and cause of insulin resistance the molecular mechanisms that trigger increased oxidant production in mitochondria as well as the precise source of these oxidants remain unclear.

In the present study, we have performed global analysis of the proteome and transcriptome in insulin-resistant adipose tissue from mice and humans and in a range of insulin resistance models in cultured adipocyte models including hyperinsulinaemia, inflammation, and glucocorticoids in an

effort to identify changes that may contribute to mitochondrial oxidant production. The mevalonate/coenzyme Q (CoQ) biosynthesis pathway was altered in all models, and this was accompanied by a selective decrease in mitochondrial CoQ content in all models of adipocyte insulin resistance as well as in human adipose tissue and in insulin-resistant muscle from mice fed a high fat high sucrose diet. Loss of mitochondrial CoQ was both necessary and sufficient to drive complex II-dependent mitochondrial oxidant production and adipocyte insulin resistance. Our data provide evidence for decreased mitochondrial CoQ content and resultant generation of oxidants being a convergent pathway for many different models of insulin resistance.

## Results

### Insulin resistance models

To identify pathways that may contribute to insulin resistance in adipose tissue we used unbiased proteomics to specifically look for factors or pathways that: (a) change across a range of insulin-resistant models including in humans, and (b) that have a demonstrable link to mitochondrial redox homeostasis. The models studied included adipose tissue from mice fed a high fat high sucrose diet (HFHSD) for different periods of time and three in vitro models of insulin resistance (3T3-L1 adipocytes treated with chronic insulin, dexamethasone or tumour necrosis factor-$\alpha$). We initially focussed on adipose tissue as our tissue of interest for two main reasons. First, adipose tissue insulin resistance is found in mice and humans that display whole body insulin resistance and insulin resistance at this site can influence whole body insulin sensitivity (*Abel et al., 2001*; *Sugii et al., 2009*). Second, there are highly robust in vitro adipocyte models (3T3-L1 cells) that accurately recapitulate both insulin action and the generation of insulin resistance using a range of insults that mimic perturbations implicated in insulin resistance in vivo such as hyperinsulinemia, inflammation and glucocorticoids. These models are invaluable since they provide a highly controlled system for manipulating insulin sensitivity in a cell autonomous manner.

Mice fed a HFHSD for one day were glucose intolerant and the extent of glucose intolerance plateaued by 14 d (*Figure 1—figure supplement 1A–B*). Adipose tissue insulin resistance was observed by 5 d and this degree of resistance was maintained to 42 d of HFHSD feeding (*Figure 1A*, *Figure 1—figure supplement 1C*). Skeletal muscle exhibited a comparatively delayed onset of insulin resistance consistent with previous studies (*Turner et al., 2013*), but also reached a maximal observed insulin resistance by 14 d (*Figure 1—figure supplement 1D*). Insulin resistance in in vitro models was defined by impaired HA-GLUT4 translocation to the plasma membrane (*Figure 1B*) and insulin-mediated 2-deoxyglucose (2DOG) uptake (*Figure 1—figure supplement 1E*). Together, these models provided an ideal integrated platform with which to explore drivers of insulin resistance.

### Proteomic analysis of insulin resistance

Proteomic analyses of insulin-resistant adipose tissue and 3T3-L1 adipocytes provided quantitative data on 2981 and 3494 proteins, respectively (*Figure 1C–D*, *Supplementary file 1*-tabs A, C), 98 of which were altered at both 5 and 14 d HFHSD time points (*Figure 1—figure supplement 1F*, top right panel, *Supplementary file 1*- tab A) and 491 in ≥2 in vitro models (*Figure 1—figure supplement 1F*, left panel, *Supplementary file 1*-tab C). A small subset of these (19) were common to both analyses (*Figure 1—figure supplement 1F*, bottom right panel, *Figure 1—figure supplement 1.L* ) including the LRP1 chaperone LRPAP1, which is of interest because LRP1 regulates GLUT4 trafficking in adipocytes (*Jedrychowski et al., 2010*). From gene set enrichment analysis, 13 pathways were altered at both time points in adipose tissue from HFHSD mice (*Figure 1—figure supplement 1G*, top right panel, *Supplementary file 1*-tab E). Similarly, there was a high degree of overlap in altered pathways across different in vitro models (*Figure 1—figure supplement 1G*, left panel, *Supplementary file 1*-tab E). Ten pathways were overlapping between all models (*Figure 1—figure supplement 1G*, bottom right panel, *Figure 1—figure supplement 1.M*). Intriguingly, parallel analysis of gene expression in these models revealed minimal overlap between regulated transcripts and proteins in all models (*Figure 1—figure supplement 1I–K*, *Supplementary file 2*). For example, of the 98 proteins altered at both 5 and 14 d HFHSD only 12 were altered at the mRNA level (*Figure 1—figure supplement 1H*, *Supplementary file 2*-tab A), and there was limited concordance

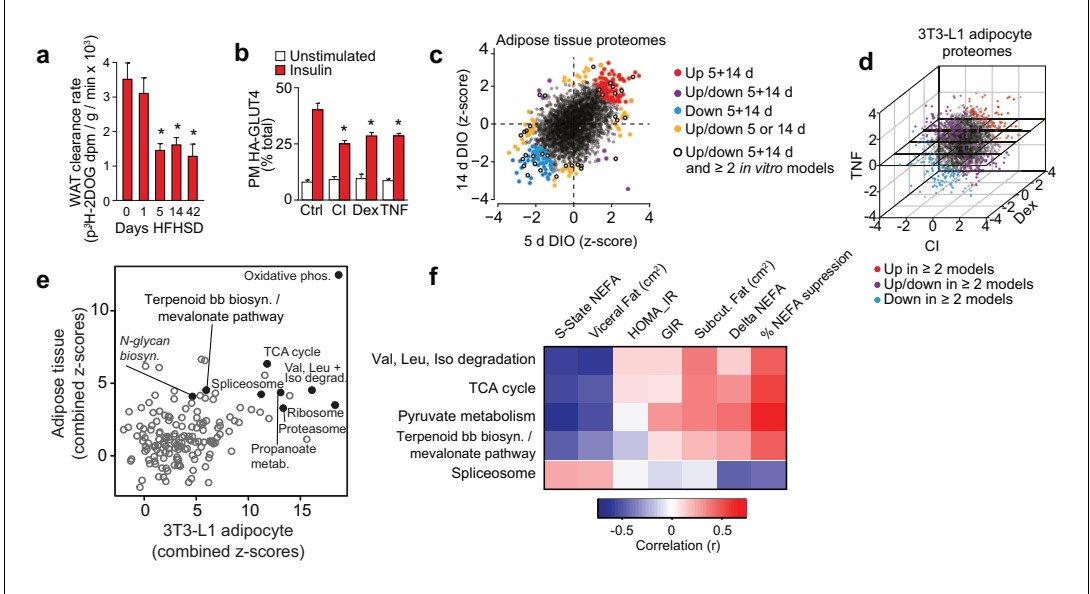

**Figure 1.** Proteomic analysis of adipocyte insulin resistance. (A) Adipose tissue [3]H-2-DOG uptake during a glucose tolerance test in mice fed a high fat high sucrose diet (HFHSD) for indicated times. Results show mean ±S.E.M. of eight mice. *p<0.05 versus mice fed a HFHSD for 0 days, t-tests corrected for multiple comparisons. (B) 3T3-L1 adipocytes were treated with chronic insulin (CI), dexamethasone (Dex) or tumour necrosis factor-α (TNF) to induce insulin resistance and stimulated with 100 nM insulin for 20 min where indicated before determination of insulin sensitivity by HA-GLUT4 abundance at the plasma membrane (PM). Results show mean ±S.E.M. of six separate experiments. *p<0.05 versus insulin-stimulated control cells, t-tests corrected for multiple comparisons. (C) Scatter plot of z-scores of protein changes (p<0.01) in adipose tissue from mice fed a HFHSD for 5 or 14 d (red = proteins consistently up-regulated, blue = proteins consistently down-regulated, purple = proteins with mixed response (up/down), orange = proteins altered at a single time point, black open circle = changed at both HFHSD time points and at least two in vitro models). (D) Three-dimensional direction analysis for proteomic data across the three 3T3-L1 adipocyte models of insulin resistance (coloured proteins = p < 0.01). Axes correspond to the z-scores of protein changes and coloured points indicate proteins changed in at least two out of three models (red = proteins up-regulated, blue = proteins down-regulated, purple = proteins with mixed response (up/down) across models). (E) Scatter plot of z-scores for pathways from proteomic analysis of insulin-resistant tissue (y-axis) or cells (x-axis). Selected pathways with z-scores >4 in tissue and cells are highlighted. (F) Heat map of correlations between expression of proteins within selected KEGG pathways and clinical measures from a cohort of obese subjects. Pathways of interest from (E) with significant (>0.423 or <−0.423) correlation with % suppression of circulating non-esterified fatty acids (NEFA) during a hyperinsulinaemic-euglycaemic clamp are shown (S-State NEFA, steady-state non-esterified free fatty acids during hyperinsulinaemic-euglycaemic clamp; HOMA_IR, Homeostatic model assessment of insulin resistance; GIR, glucose infusion rate during a hyperinsulinaemic-euglycaemic clamp; Delta NEFA, change in circulating NEFA concentrations between baseline and during a hyperinsulinaemic-euglycaemic clamp; % suppression NEFA, percentage suppression of NEFAs between baseline and during a hyperinsulinaemic-euglycaemic clamp; Val, valine; Leu, leucine; Iso, isoleucine; TCA, tricarboxylic acid cycle; Terpenoid bb biosyn., Terpenoid backbone biosynthesis). See also *Figure 1—figure supplement 1*.

DOI: https://doi.org/10.7554/eLife.32111.003

The following figure supplement is available for figure 1:

**Figure supplement 1.** Comparison of transcriptomic and proteomic changes in insulin-resistant adipocytes.
DOI: https://doi.org/10.7554/eLife.32111.004

between changes in proteins and transcript expression for the 19 proteins and 10 pathways found to be altered in both in vivo and in vitro models (*Figure 1—figure supplement 1L–M*). This lack of concordance between gene and protein expression emphasises how crucial proteomic analyses are in identifying causal links to metabolic disease.

## Integrated analysis of insulin-resistant proteomes

To identify convergent changes in pathways at the proteome-level that correlated with insulin resistance across all adipocyte models, we generated a combined z-score for pathways across in vivo time points and in vitro models (*Figure 1E*). There were 13 KEGG pathways (excluding disease pathways) that were highly altered (z-score >4) in in vivo and in vitro analyses. *Oxidative phosphorylation* was most highly altered in both in vivo and in vitro models, and other pathways of interest included *TCA cycle*, branched chain amino acid metabolism (*valine, leucine and isoleucine degradation*),

*proteasome*, *ribosome*, *spliceosome*, *N-glycan biosynthesis* and *terpenoid backbone biosynthesis/mevalonate pathway* (*Figure 1E*, *Supplementary file 3*- tab B).

## Proteomic analysis of human adipose insulin resistance

To further filter pathways that might be implicated in insulin resistance, we next performed proteomic analysis of adipose tissue from a cohort of obese subjects that have been extensively clinically phenotyped (*Chen et al., 2015*). This cohort was matched for BMI and comprised insulin- sensitive and insulin-resistant subjects based on responses during a hyperinsulinaemic-euglycaemic clamp, meaning that we could identity pathways related to insulin sensitivity independent of obesity/BMI (*Chen et al., 2015*). We quantified 4481 proteins across 22 subjects and correlated the expression of proteins (*Supplementary file 3*- tab A) and pathways (*Supplementary file 3*- tab B) with clinical features that are diagnostic of insulin sensitivity. For the purposes of this exercise, we focused on suppression of non-esterified fatty acids (NEFAs) during the clamp as this is likely to be more directly related to insulin action in adipose tissue than glucose infusion rate (GIR), which is likely driven mainly by muscle. We identified 299 proteins (*Supplementary file 3*- tab A) and 26 pathways (*Supplementary file 3*- tab B) that were positively correlated with insulin sensitivity and 142 proteins and two pathways (*ribosome*, *spliceosome*) that were negatively correlated with insulin sensitivity (r = <−0.423 or >0.423) (*Supplementary file 3*- tabs A, B). Importantly, the *PPAR signalling* pathway, a known regulator of adipose insulin sensitivity (*Sugii et al., 2009*), was positively associated with insulin sensitivity in this analysis. Of the 13 pathways of interest from the integrated proteomic analysis of insulin resistance models (*Figure 1E*) only five were positively associated with insulin sensitivity in human adipose tissue (*Figure 1F*, *Supplementary file 3*-tab B). These comprised *spliceosome*, central carbon metabolism (*pyruvate metabolism, TCA cycle, glycolysis, pentose phosphate pathway*), amino acid metabolism including *branched chain amino acid synthesis/ degradation* and the *terpenoid backbone biosynthesis/mevalonate pathway*, which generates precursors for isoprenoids such as cholesterol and CoQ. This is of interest as branched chain amino acid metabolism (*Newgard et al., 2009*) and spliceosome function (*Vernia et al., 2016*) have been implicated in adipocyte or whole body insulin sensitivity, providing support for our analysis pipeline.

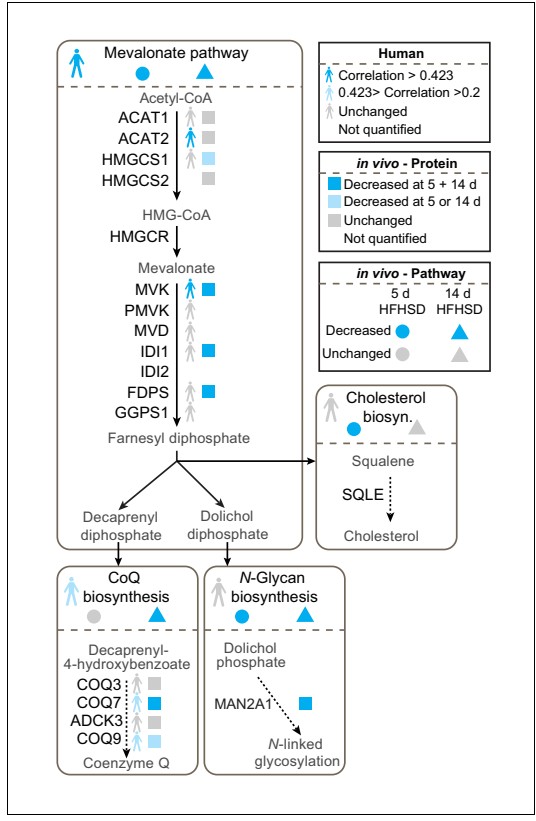

**Figure 2.** Proteomic data for the mevalonate pathway and downstream pathways in insulin-resistant adipose tissue. Human figures represent expression of pathways or proteins in human data (*Supplementary file 3*). Circles and triangles represent gene set enrichment pathway analyses for proteomic analysis of tissue from mice fed a HFHSD for 5 (circle) or 14 d (triangle) as indicated in the figure legend (significance = adj. p<0.05). Squares represent individual protein data (adj. p<0.01). Statistical analyses described in Materials and Methods. All proteins in mevalonate/terpenoid backbone biosynthesis pathway are depicted along with selected proteins from CoQ biosynthesis, *N*-glycan biosynthesis and cholesterol/steroid biosynthesis. Colours indicate direction of change as in legend. See also *Figure 2—figure supplement 1*.
DOI: https://doi.org/10.7554/eLife.32111.005

The following figure supplement is available for figure 2:

**Figure supplement 1.** Pathway and single gene/protein data for the mevalonate pathway and downstream pathways in insulin-resistant adipose tissue and in vitro models of insulin resistance.
DOI: https://doi.org/10.7554/eLife.32111.006

## The mevalonate and coenzyme Q biosynthesis pathways are altered in insulin resistance

We were particularly interested in the potential role of the mevalonate pathway in insulin resistance because this pathway feeds CoQ biosynthesis, an essential component of mitochondrial electron transport and so defects in this pathway might play a role in mitochondrial oxidative stress. Our data revealed changes in transcripts and proteins throughout the mevalonate pathway (*Figure 2*, *Figure 2—figure supplement 1*, *Supplementary file 1*- tabs A-E, *Supplementary file 2*- tabs A-E, *Supplementary file 3*- tabs A-B), but when assessing different endpoints of the mevalonate pathway such as cholesterol, *N*-glycosylation (dolichol) and CoQ, we only observed altered protein expression in components of the CoQ pathway in both human and mouse adipose tissue (*Figure 2*). COQ7 and COQ9 were lower in insulin-resistant adipose tissue from humans and mice (*Figure 2*), while ADCK3/COQ8 was dysregulated at the protein and mRNA levels in vitro (*Figure 2—figure supplement 1*). Our integrated analysis of insulin-resistant proteomes from in vivo and in vitro models, and human, pointed toward a convergence upon dysregulated CoQ biosynthesis in insulin resistance.

## Mitochondrial CoQ content is decreased selectively in insulin resistance

We next determined if the change in expression of mevalonate/CoQ pathway proteins translated into altered CoQ metabolism. To do this, we measured total CoQ content in in vivo and in vitro models of adipocyte insulin resistance. Whole-cell CoQ concentrations were decreased in in vitro models (*Figure 3A*) but not in insulin-resistant adipose tissue (*Figure 3B*). We postulated that because CoQ is found in membranes throughout the cell that there might be a selective depletion of CoQ in specific locations, for example in mitochondria where it is synthesised. To investigate this, we analysed CoQ in subcellular fractions from 3T3-L1 adipocytes and adipose tissue. This revealed a selective depletion of CoQ in mitochondria across all models (*Figure 3C–D*, *Figure 3—figure supplement 1A–B*). This decrease was not due to changes in mitochondrial content (assessed via citrate synthase activity and OXPHOS subunit abundance; *Figure 3—figure supplement 1C–J*, *Supplementary file 2*- tabs A-D). Consistent with data from model systems, adipose tissue mitochondrial $CoQ_{10}$ (the major form of CoQ in humans) was positively correlated with insulin-induced suppression of NEFAs (*Figure 3E–F*, *Table 1*) and whole-body insulin sensitivity (GIR) in obese humans (*Figure 3—figure supplement 1K–L*, *Table 1*). Mitochondrial $CoQ_{10}$ was also significantly and positively correlated with expression of proteins in the CoQ biosynthesis pathway (r =+0.52) (*Supplementary file 3*-tab B) indicating that decreased biosynthesis may contribute to lower mitochondrial CoQ in insulin resistance. Stratification of participants by adipose tissue mitochondrial $CoQ_{10}$ revealed no effect of age or BMI (*Table 1*). These findings reveal that decreases in mitochondrial CoQ are an obesity-independent feature of adipocyte insulin resistance.

Intriguingly, our proteomic data indicated that the expression of proteins integral to the mevalonate pathway was decreased in fat from humans and mice and from 3T3-L1 adipocytes treated with dexamethasone or TNF-α whereas this was not the case in the chronic insulin 3T3-L1 adipocyte model (*Figure 2—figure supplement 1*). Thus, we next examined if the observed decrease in mitochondrial CoQ reflected changes in CoQ biosynthesis, which we measured by determining $^{13}C_6$-$CoQ_9$ in 3T3-L1 adipocytes incubated with $^{13}C_6$-4-hydroxybenzoic acid. Consistent with pathway analysis and our intracellular measures of cholesterol content (*Figure 3—figure supplement 1M–P*), CoQ biosynthesis rates were lower in cells treated with dexamethasone or TNF-α but elevated in response to chronic insulin (*Figure 3—figure supplement 1Q*). Together, it appears probable that dexamethasone and TNF-α treatments lower mitochondrial CoQ largely via reduced biosynthesis, although increased CoQ in microsomal and PM subcellular fractions (*Figure 3—figure supplement 1A–B*) in these models point to additional dysregulation of CoQ trafficking. Since these models replicate the lower content of mevalonate/CoQ biosynthesis pathway proteins measured in mice and humans, it is likely that decreased CoQ biosynthesis contributes to loss of CoQ in these more physiological systems. This does not appear to be the case for adipocytes treated with chronic insulin, where additional pathway(s) likely contribute to dysregulated mitochondrial CoQ homeostasis.

The above findings highlight loss of mitochondrial CoQ as a common feature of adipocyte insulin resistance so we next investigated if a similar phenomenon occurs in other insulin responsive tissues, most notably muscle in view of its major role in whole body glucose metabolism/insulin resistance. In muscle, we found decreased mitochondrial CoQ at 14 and 42 d HFHSD feeding (*Figure 3G*). These

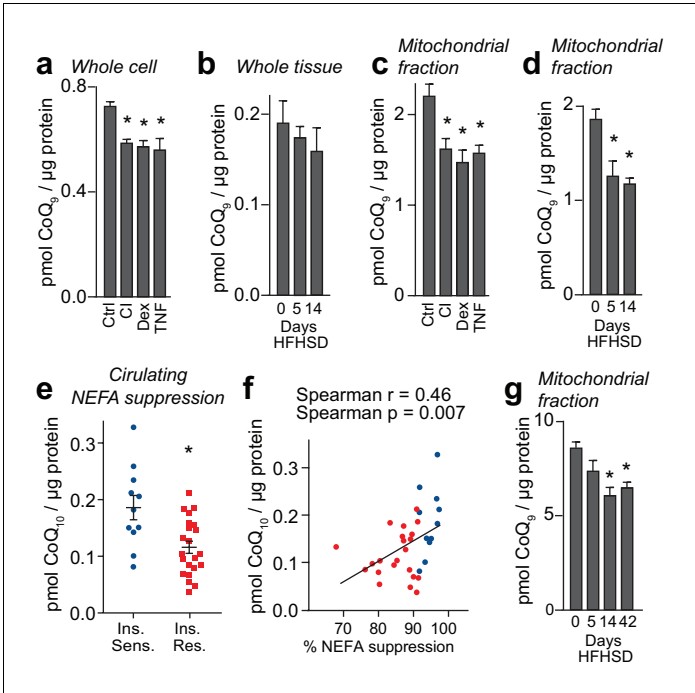

**Figure 3.** Loss of mitochondrial coenzyme Q in insulin-resistant adipose and muscle tissue. (A) Concentrations of $CoQ_9$ in control (Ctrl) 3T3-L1 adipocytes, and adipocytes treated to induce insulin resistance with chronic insulin treatment (CI), dexamethasone (Dex) or TNFα (TNF). (B) Total adipose tissue concentrations of $CoQ_9$ were determined in mice fed a HFHSD for 0, 5 and 14 d. (C) Concentrations of $CoQ_9$ in the mitochondria of 3T3-L1 adipocytes treated as specified. (D) Concentrations of $CoQ_9$ in mitochondria of adipose tissue from mice fed a HFHSD for indicated times. (A–D) Results show mean ±S.E.M of three to five separate in vitro studies and six in vivo studies. *$p < 0.05$ versus control samples, t-tests corrected for multiple comparisons. (E) Obese participants were stratified into insulin-sensitive (upper tertile, 11 subjects) and insulin-resistant (lower two tertiles, 22 subjects) groups based on the % reduction in circulating NEFAs during a hyperinsulinaemic-euglycaemic clamp. Mean adipose mitochondrial $CoQ_{10}$ concentrations were calculated for each group, error bars are S.E.M. (Mann-Whitney, *$p < 0.05$). (F) Correlation between rate of reduction (%) in circulating NEFA and adipose mitochondrial $CoQ_{10}$ concentrations for human subjects in *Figure 3E*. Colours correspond to colours in *Figure 3E*. (G) Concentrations of $CoQ_9$ in mitochondria of skeletal muscle (quadriceps) from mice fed a HFHSD for indicated times. Results show mean ±S.E.M of four to six in vivo studies. *$p < 0.05$ versus control samples, t-tests corrected for multiple comparisons. See also *Figure 3—figure supplement 1*.

DOI: https://doi.org/10.7554/eLife.32111.007

The following figure supplement is available for figure 3:

**Figure supplement 1.** CoQ, cholesterol and mitochondria content in in vitro and in vivo models of insulin resistance.

DOI: https://doi.org/10.7554/eLife.32111.008

time points correlate with the emergence of insulin resistance in muscle (*Figure 1—figure supplement 1D*). In contrast to adipose tissue, total muscle CoQ was lower at all time points tested (*Figure 3—figure supplement 1R*), potentially reflecting higher mitochondrial content of this tissue. Liver mitochondrial CoQ was unchanged in response to HFHSD feeding (*Figure 3—figure supplement 1S*), despite changes in total CoQ at 5 d HFHSD feeding (*Figure 3—figure supplement 1T*). Changes in cholesterol content in in vitro models (*Figure 3—figure supplement 1M–P*), adipose tissue (*Figure 3—figure supplement 1U*), muscle (*Figure 3—figure supplement 1V*) and liver (*Figure 3—figure supplement 1W*) were inconsistent with a causal role in insulin resistance across multiple tissues. These data suggest that a decrease in mitochondrial CoQ may be involved at an early stage in the development of insulin resistance in muscle and adipose tissue.

**Table 1.** Anthropometric, clinical and metabolic characteristics of obese females stratified into a upper tertile and lower two tertiles based on adipose tissue mitochondrial CoQ content (CoQ$_{high}$ vs CoQ$_{low}$).
All p values calculated by Mann Whitney test, p>0.05 in bold.

| Characteristics | CoQ$_{high}$ (n = 11) | CoQ$_{low}$ (n = 22) | P value |
|---|---|---|---|
| Age, y | 53 ± 11 | 51 ± 13 | 0.70 |
| BMI, kg/m$^2$ | 36.6 ± 4.2 | 36.9 ± 5.1 | 0.92 |
| Subcutaneous fat, cm$^2$ a | 588 ± 132 | 546 ± 114 | 0.33 |
| Visceral fat, cm$^2$ a | 210 ± 37 | 264 ± 74 | 0.07 |
| Liver fat, % a | 6.8 ± 4.5 | 13.6 ± 10.3 | 0.06 |
| Mean adipocyte size, μm b | 71.3 ± 8.9 | 72.6 ± 9.0 | 0.90 |
| Total cholesterol, mmol/L | 5.0 ± 0.8 | 4.8 ± 0.7 | 0.87 |
| LDL cholesterol, mmol/L | 3.0 ± 0.8 | 2.9 ± 0.6 | 0.89 |
| HDL cholesterol, mmol/L | 1.5 ± 0.2 | 1.4 ± 0.3 | 0.19 |
| Triacylglycerides, mmol/L | 0.8 ± 0.3 | 1.0 ± 0.4 | 0.15 |
| Fasting insulin, mU/L | 14.2 ± 4.9 | 16.0 ± 7.4 | 0.71 |
| Fasting NEFA, mmol/L | 0.46 ± 0.14 | 0.43 ± 0.13 | 0.63 |
| NEFA during low dose insulin infusion, mmol/L | 0.03 ± 0.02 | 0.06 ± 0.03 | **0.03** |
| NEFA suppression during low dose insulin infusion, % | 92.1 ± 4.5 | 86.5 ± 6.9 | **0.01** |
| Glucose infusion rate during high dose insulin infusion, μmol/kg fat free mass/min | 117.8 ± 26.6 | 98.5 ± 29.4 | **0.02** |
| Endogenous glucose production suppression during low-dose insulin infusion, % | 73.6 ± 12.1 | 60.5 ± 15.7 | **0.03** |
| HOMA-IR score | 2.9 ± 1.1 | 3.5 ± 1.7 | 0.40 |

a –CoQ$_{high}$n = 10, CoQ$_{low}$n = 22.
b - CoQ$_{high}$n = 9, CoQ$_{low}$n = 18.
DOI: https://doi.org/10.7554/eLife.32111.009

## Supplementation of CoQ restores mitochondrial CoQ and insulin sensitivity

To test whether loss of CoQ contributes to insulin resistance in adipocytes and muscle, we first examined whether restoration of mitochondrial CoQ could restore insulin sensitivity in 3T3-L1 adipocytes. Addition of CoQ$_9$ had no effect in control cells but increased mitochondrial CoQ$_9$ in insulin-resistant cells (*Figure 4A*) and both CoQ$_9$ and CoQ$_{10}$ improved insulin-stimulated HA-GLUT4 translocation and 2DOG uptake in all cell models (*Figure 4B*, *Figure 4—figure supplement 1A–B*). In light of the importance of adipocyte lipolysis in whole body glucose homeostasis (*Perry et al., 2015*; *Titchenell et al., 2016*), we next tested whether CoQ could also improve insulin-regulated suppression of lipolysis in these models. Dex or TNFα treatment enhanced basal lipolysis as previously described (*Souza et al., 1998*; *Xu et al., 2009*) and insulin-regulated inhibition of lipolysis was defective in all models (*Figure 4C*, *Figure 4—figure supplement 1C–E*). Provision of CoQ$_9$ had no effect in control cells but increased suppression of lipolysis by insulin in in vitro models (albeit not significantly in the TNF model; p=0.11).

We next tested whether CoQ could alleviate insulin resistance in vivo by providing liposomal CoQ$_{10}$ via intraperitoneal injection every second day. CoQ$_{10}$ was used for these studies as it was not feasible to obtain sufficient CoQ$_9$ and the doses of CoQ$_{10}$ used in these studies were optimised so that we did not observe changes in body weight or adiposity, reported previously (*Xu et al., 2017*), since such metabolic changes would likely directly affect insulin action in muscle and adipose. CoQ$_{10}$ administration improved whole-body glucose tolerance in mice fed a HFHSD for 5 or 14 d (*Figure 4D–E*, *Figure 4—figure supplement 1F–G*) without altering insulin secretion (*Figure 4F*, *Figure 4—figure supplement 1H*). Improved glucose tolerance was accompanied by increased suppression of NEFAs (*Figure 4G*, *Figure 4—figure supplement 1I*) and 2DOG clearance into epididymal (*Figure 4—figure supplement 1J*) and inguinal (*Figure 4—figure supplement 1K*, p=0.06) adipose depots and quadriceps (*Figure 4—figure supplement 1L*, p=0.08) during the GTT. In a

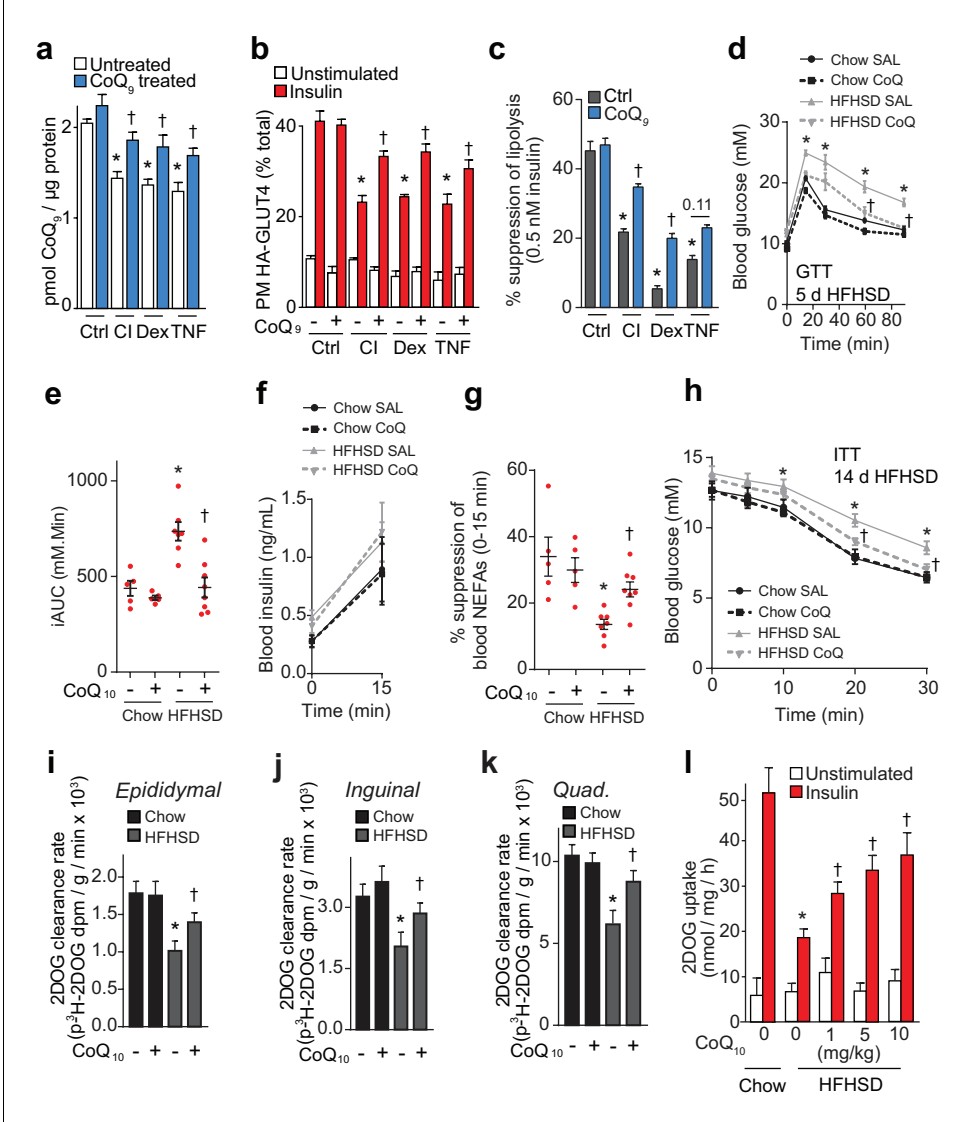

**Figure 4.** CoQ supplementation improves insulin sensitivity in vitro and in vivo models of insulin resistance. (A) Concentrations of CoQ$_9$ in mitochondria of control (Ctrl) and insulin-resistant 3T3-L1 adipocyte models supplemented with 10 µM CoQ$_9$ as indicated. Results show mean ±S.E.M. of three to four independent experiments. *p<0.05 versus control cells, †p<0.05 versus respective cells without CoQ supplementation, t-tests corrected for multiple comparisons. (B–C) 3T3-L1 adipocytes were treated to induce insulin resistance with and without 10 µM CoQ$_9$ supplementation as indicated before determination of insulin sensitivity by HA-GLUT4 abundance at the plasma membrane (PM) in unstimulated cells and in response to 100 nM insulin (B) or by suppression of lipolysis by 0.5 nM insulin (C; calculated from *Figure 4—figure supplement 1C–E* as described in the Materials and Methods). Results show mean ±S.E.M. of three to four independent experiments. *p<0.05 versus insulin-stimulated control cells, †p<0.05 versus respective insulin-stimulated cells without CoQ supplementation, t-tests corrected for multiple comparisons. (D) Blood glucose concentrations during a glucose tolerance test (GTT) in mice treated as indicated (5 d HFHSD; saline (SAL) or 10 mg/kg CoQ$_{10}$). Results show mean ±S.E.M. of five to nine mice. *p<0.05 versus chow-fed saline treated mice at the same time-point, †p<0.05 versus HFHSD-fed saline-treated mice at the same time point, t-tests corrected for multiple comparisons. (E) Incremental area under the curve (iAUC) for blood concentrations during GTT in *Figure 4D*. *p<0.05 versus chow-fed saline treated mice, †p<0.05 versus HFHSD-fed saline-treated mice, t-tests corrected for multiple comparisons. (F) Blood insulin concentrations at 0 and 15 min of the GTT in *Figure 4D*. (G) Suppression of circulating non-esterified fatty acids (NEFAs) between 0 and 15 min of the GTT in *Figure 4D*. *p<0.05 versus chow-fed saline treated mice, †p<0.05 versus HFHSD-fed saline-treated mice, t-tests corrected for multiple comparisons. (H) Blood glucose concentrations during an ITT in mice treated as indicated (14 d HFHSD; 10 mg/kg CoQ$_{10}$). Results show mean ±S.

*Figure 4 continued on next page*

*Figure 4 continued*

E.M. of five to nine mice. *p<0.05 versus chow-fed saline treated mice at the same time-point, †p<0.05 versus HFHSD-fed saline-treated mice at the same time point, t-tests corrected for multiple comparisons. (I–K) Epididymal (I) and inguinal (J) adipose tissue and quadriceps (K) $^3$H-2-DOG uptake during the ITT in *Figure 4H*. Results show mean ±S.E.M. of five to nine mice. *p<0.05 versus chow-fed saline treated mice at the same time-point, †p<0.05 versus HFHSD-fed saline-treated mice at the same time point, t-tests corrected for multiple comparisons. (L) Adipose tissue explants from mice fed a chow diet or a HFHSD for 14 d and supplemented with CoQ$_{10}$ at specified doses were stimulated with 10 nM insulin where indicated and 2-DOG uptake was assessed. Results show mean ±S.E.M. of five mice. *p<0.05 versus insulin-stimulated explants from mice fed a chow diet and †p<0.05 versus insulin-stimulated explants from HFHSD-fed mice without CoQ supplementation, t-tests corrected for multiple comparisons. See also *Figure 4—figure supplement 1*.

DOI: https://doi.org/10.7554/eLife.32111.010

The following figure supplement is available for figure 4:

**Figure supplement 1.** Addtional data on the effect of CoQ supplementation on insulin sensitivity in models of insulin resistance

DOI: https://doi.org/10.7554/eLife.32111.011

---

more direct measure of insulin sensitivity in vivo, CoQ$_{10}$ also increased insulin responsiveness of HFHSD-fed mice during an ITT as measured by blood glucose excursion (*Figure 4H*) and 2DOG uptake into epididymal and inguinal fat pads and quadriceps muscle (*Figure 4I–K*).

Adipose depots retain a high degree of insulin responsiveness and insulin resistance ex vivo (*Figure 1—figure supplement 1C*, *Figure 4L*). Therefore, we used adipose tissue for ex vivo analyses rather than muscle tissue, for which ex vivo analyses are more technically challenging and give less robust responses (data not shown). Administration of increasing doses of CoQ$_{10}$ to HFHSD-fed mice for 14 d dose-dependently increased CoQ$_{10}$ in mitochondria from epididymal adipose tissue (*Figure 4—figure supplement 1M*), improved glucose tolerance (*Figure 4—figure supplement 1N–O*) and improved insulin-stimulated 2DOG uptake in an ex vivo assay using adipose tissue explants (*Figure 4L*). CoQ$_{10}$ did not alter whole body or epididymal fat pad mass (*Figure 4—figure supplement 1P–Q*). Together, these data suggest that provision of exogenous CoQ improves insulin-regulated glucose uptake and that lower CoQ content may contribute to insulin resistance in adipose tissue and muscle.

## Inhibition of CoQ synthesis induces insulin resistance

Our previous data shows that lower mitochondrial CoQ content is a common feature of insulin resistance and that replacing CoQ can overcome insulin resistance. We next investigated whether specific perturbation of CoQ biosynthesis was sufficient to induce insulin resistance in adipocytes. To achieve this, we incubated adipocytes with 4-nitrobenzoic acid (NB) or 4-cholorobenzoic acid (CB) (*Alam et al., 1975*; *Forsman et al., 2010*) to competitively inhibit 4-hydroxybenzoate:polyprenyl transferase (Coq2) (*Figure 5A*). Both inhibitors decreased mitochondrial CoQ$_9$ (*Figure 5A*) to a similar extent to that observed in insulin-resistant adipocytes (*Figure 3C*), while CoQ$_9$ supplementation restored normal mitochondrial CoQ$_9$ concentrations (*Figure 5A*). Importantly, both inhibitors caused insulin resistance in adipocytes (*Figure 5B–5C*, *Figure 5—figure supplement 1A–B*) which could be reversed with provision of CoQ. Similarly, siRNA knock down of the key regulatory proteins in CoQ biosynthesis that were down-regulated in insulin-resistant mouse and human adipose tissue (*Coq7* or *Coq9*) (*Figure 5—figure supplement 1C–D*) lowered mitochondrial CoQ$_9$ (*Figure 5D*) and triggered insulin resistance (*Figure 5E–5F*, *Figure 5—figure supplement 1E*). Insulin resistance triggered by pharmacological or genetic inhibition of CoQ biosynthesis occurred independently of consistent defects in insulin signalling to the key regulators of glucose transport (Akt or the Akt substrate TBC1D4), or changes in GLUT4 expression (*Figure 5—figure supplement 1F–M*). Similarly, CoQ provision did not alter signalling responses or GLUT4 expression in any in vitro models of insulin resistance (*Figure 5—figure supplement 1N–Q*) despite improved insulin-stimulated HA-GLUT4 translocation, 2DOG uptake and suppression of lipolysis under these conditions (*Figure 4B–C*, *Figure 4—figure supplement 1A*). These data are consistent with previous reports that insulin resistance is not driven by overt and consistent defects in proximal insulin signalling (*Hoehn et al., 2008*; *Tan et al., 2015*).

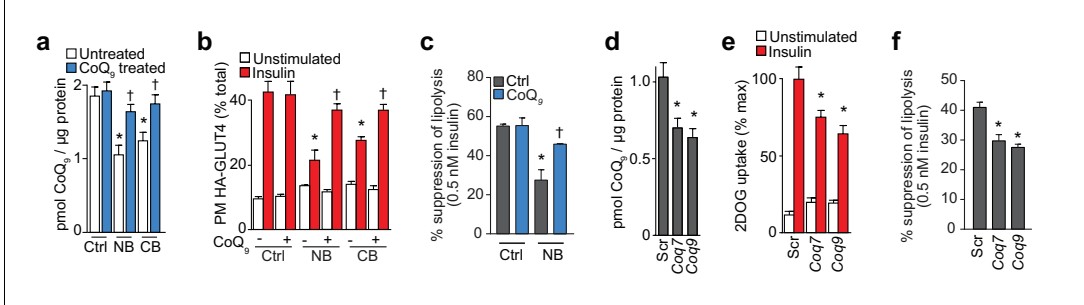

**Figure 5.** Inhibition of CoQ synthesis is sufficient for insulin resistance in vitro. (**A**) Concentrations of CoQ$_9$ in mitochondria of control (Ctrl) 3T3-L1 adipocytes and adipocytes treated with 2.5 mM 4-nitrobenzoic (NB) or 4-chlorobenzoic acid (CB) with or without 10 μM CoQ$_9$. Results show mean ±S.E. M. for five-six separate in vitro experiments. *p<0.05 versus control cells, †p<0.05 versus respective model without CoQ supplementation, t-tests corrected for multiple comparisons. (**B–C**) 3T3-L1 adipocytes were treated with NB or CB before sensitivity was determined by HA-GLUT4 abundance at the plasma membrane (PM) before and after stimulation with 100 nM insulin (**B**) or suppression of lipolysis by 0.5 nM insulin (**C**; calculated from *Figure 5—figure supplement 1B* as described in the Materials and Methods). Results show mean ±S.E.M. for four separate in vitro experiments. *p<0.05 versus insulin-stimulated control cells, †p<0.05 versus respective insulin-stimulated cells without CoQ supplementation, t-tests corrected for multiple comparisons. (**D**) Concentrations of CoQ$_9$ in mitochondria of 3T3-L1 adipocytes treated with control scrambled (Scr) siRNA, or pooled siRNA targeted to *Coq7* or *Coq9*. Results show mean ±S.E.M. for four separate in vitro experiments. *p<0.05 versus control cells, t-tests corrected for multiple comparisons. (**E–F**) Insulin sensitivity of 3T3-L1 adipocytes treated with Scr or siRNA targeted to *Coq7* or *Coq9* as measured by 2DOG uptake (**E**) or suppression of lipolysis by 0.5 nM insulin (**F**; calculated from *Figure 5—figure supplement 1E* as described in the Materials and Methods). Results show mean ±S.E.M. for four separate in vitro experiments. *p<0.05 versus insulin-stimulated control cells, t-tests corrected for multiple comparisons. See also *Figure 5—figure supplement 1*.

DOI: https://doi.org/10.7554/eLife.32111.012

The following figure supplement is available for figure 5:

**Figure supplement 1.** Loss of CoQ causes insulin resistance independently of consistent defects in insulin signalling or changes in GLUT4 expression.
DOI: https://doi.org/10.7554/eLife.32111.013

The link between CoQ and insulin resistance is of interest in the context of statins that target the mevalonate pathway and have recently been shown to be associated with progression to type two diabetes in humans (*Cederberg et al., 2015*; *Preiss et al., 2011*; *Sattar et al., 2010*). To begin to explore this, we incubated 3T3-L1 adipocytes with simvastatin or atorvastatin for up to 72 hr. Both statins lowered cellular cholesterol (*Figure 5—figure supplement 1R*) and CoQ content (*Figure 5—figure supplement 1S*), providing proof-of-principle that lower mevalonate pathway activity influences cellular CoQ content in adipocytes. Statins induced insulin resistance (*Figure 5—figure supplement 1T*), and this was reversed by providing CoQ$_9$ or mevalonate (*Figure 5—figure supplement 1T*). Together with the observation that more specific inhibitors of the CoQ biosynthetic pathway also trigger insulin resistance, these data provide convincing evidence that loss of CoQ is sufficient to induce adipocyte insulin resistance, and this may contribute to off-target effects of statin therapy.

## Loss of mitochondrial CoQ induces insulin resistance via increased mitochondrial oxidants

Mitochondrial CoQ is essential for cellular respiration, as it shuttles electrons from various membrane-bound/associated dehydrogenase complexes to complex III during oxidative phosphorylation. In addition, CoQ can regulate the formation of superoxide anion radicals from the various CoQ-interacting sites of complexes I, II and III. In this respect, excess mitochondrial CoQ above that required for maximal respiratory flux can be thought of as an 'electron sink'. Consistent with this concept, previous studies have shown that modest loss of mitochondrial CoQ can be tolerated for electron transport activity, but at the cost of increased mitochondrial oxidants (*Quinzii et al., 2008*).

We hypothesised that loss of CoQ in mitochondria may contribute to increased oxidants in insulin resistance. To test this possibility, we utilised peroxiredoxin (PRDX) dimerisation as an indicator of subcellular oxidant burden (*Perkins et al., 2015*). Peroxiredoxins undergo homodimerisation as part of their mechanism to reduce hydroperoxides particularly hydrogen peroxide (H$_2$O$_2$). Therefore, the PRDX dimer:monomer ratio is a useful surrogate to assess subcellular H$_2$O$_2$ (*Bayer et al., 2013*).

There was no change in the total content of PRDX1-3 in insulin-resistant models (*Figure 6A*, *Figure 6—figure supplement 1A,C,J*) and the dimer:monomer ratio of cytosolic PRDX1 and PRDX2 also remained unchanged (*Figure 6—figure supplement 1A–D*). In contrast, the dimer:monomer ratio of mitochondrial PRDX3 increased significantly in all in vitro models (*Figure 6A–6B*). Increased dimerisation of PRDX3 was also observed under conditions of pharmacological (*Figure 6C*, *Figure 6—figure supplement 1E*) or genetic inhibition (*Figure 6—figure supplement 1G–H*) of CoQ biosynthesis, with limited or no changes in PRDX2 redox state (*Figure 6—figure supplement 1F,I*). Restoration of normal mitochondrial $CoQ_9$ content by provision of exogenous $CoQ_9$ lowered the PRDX3 dimer:monomer ratio (*Figure 6D–6E*). PRDX3 dimerisation was also enhanced in insulin-resistant adipose tissue at 5 and 14 d HFHSD feeding (*Figure 6F*, *Figure 6—figure supplement 1J*). In vivo administration of $CoQ_{10}$ under conditions that improved insulin sensitivity lowered the PRDX3 dimer-to-monomer ratio in a dose-dependent manner (*Figure 6G*, *Figure 6—figure supplement 1K*). Together, these data place decreased mitochondrial CoQ upstream of increased mitochondrial oxidants, most likely in the form of $H_2O_2$ in adipocyte insulin resistance.

To determine if increased mitochondrial $H_2O_2$ was necessary for loss of mitochondrial CoQ to cause insulin resistance, we over-expressed mitochondria-targeted catalase (*Figure 6—figure supplement 1L*) in the setting of CoQ deficiency. This lowered the PRDX3 dimer:monomer ratio in cells

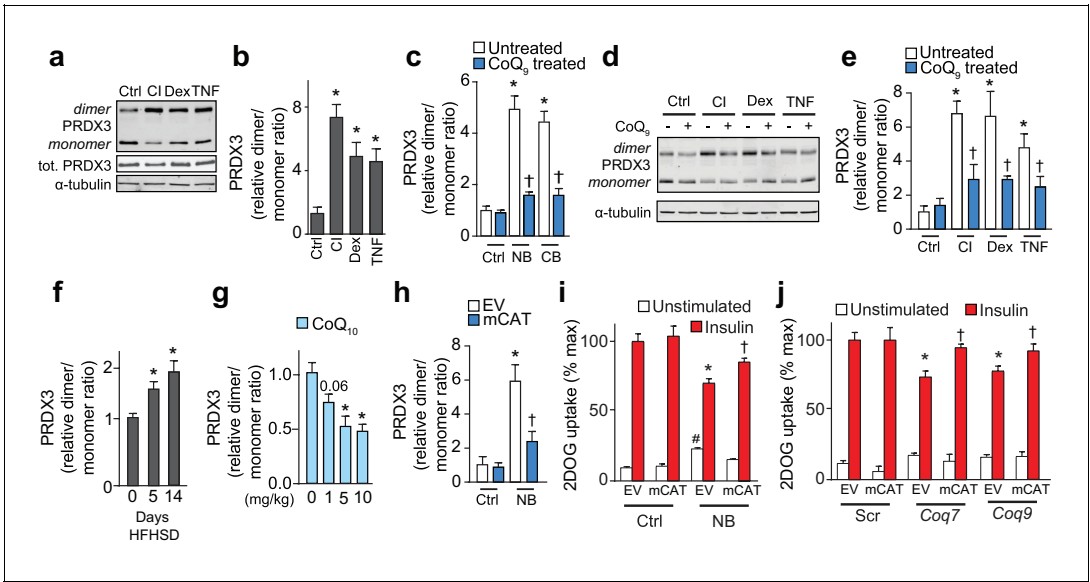

**Figure 6.** Loss of mitochondrial CoQ causes insulin resistance via mitochondrial oxidants. (A–E) PRDX3 dimerisation was assessed by immunoblot in 3T3-L1 adipocytes treated as indicated, and the PRDX3 dimer/monomer ratio was measured by densitometry and expressed relative to control cells (B, C, E). Results show mean ±S.E.M. for four to eight separate in vitro experiments. For B, *p<0.05 versus control cells, t-tests corrected for multiple comparisons. For C and E, *p<0.05 versus control cells, †p<0.05 versus respective treatment without CoQ supplementation, t-tests corrected for multiple comparisons. (F) PRDX3 dimer/monomer ratio in adipose tissue obtained from mice fed a HFHSD (relative to ratios from adipose tissue from mice fed a HFHSD for 0 d). Results show mean of six mice, *p<0.05 versus control, t-tests corrected for multiple comparisons. (G) PRDX3 dimer/monomer ratio in adipose tissue from mice fed a HFHSD for 14 d and supplemented with $CoQ_{10}$ at the specified dose. Results show mean of six mice (relative to ratios from adipose tissue from mice without $CoQ_{10}$ supplementation). *p<0.05 versus control, t-tests corrected for multiple comparisons. (H) PRDX3 dimer/monomer ratio in adipocytes expressing empty vector (EV) or mitochondria-targeted catalase (mCAT) treated with and without NB. Ratios expressed relative to control EV cells. Results show mean ±S.E.M. for four separate in vitro experiments. *p<0.05 versus untreated EV control cells, †p<0.05 versus NB-treated EV cells, t-tests corrected for multiple comparisons. (I) 3T3-L1 adipocytes expressing EV or mCAT treated were treated with NB before insulin sensitivity was determined by [3]H-2-DOG uptake. (J) [3]H-2-DOG uptake in EV or mCAT expressing 3T3-L1 adipocytes treated with scrambled siRNA (Scr) or siRNA targeted to *Coq7* or *Coq9* and stimulated with insulin where indciated. Results show mean ±S.E.M. for four separate in vitro experiments. *p<0.05 versus insulin-stimulated EV control cells, †p<0.05 versus respective insulin-stimulated EV cells, t-tests corrected for multiple comparisons.

DOI: https://doi.org/10.7554/eLife.32111.014

The following figure supplement is available for figure 6:

**Figure supplement 1.** Adipocyte insulin resistance and loss of CoQ is associated with increased oxidants specifically in mitochondria.

DOI: https://doi.org/10.7554/eLife.32111.015

where CoQ biosynthesis was inhibited (*Figure 6H Figure 6—figure supplement 1M–O*), and improved insulin responses in these conditions (*Figure 6I–6J*), consistent with loss of mitochondrial CoQ causing insulin resistance via mitochondrial $H_2O_2$.

## Loss of mitochondrial CoQ impairs insulin action via complex II-derived $H_2O_2$

We next examined the effect of loss of mitochondrial CoQ on mitochondrial function and oxidant production in more detail. Although the relationship between impaired mitochondrial function and insulin resistance is controversial (*Montgomery and Turner, 2015*), we first examined whether insulin resistance or loss of CoQ was associated with bioenergetic defects since CoQ plays a key role in oxidative phosphorylation. We assessed mitochondrial respiration in all in vitro models of insulin resistance. In control and insulin-resistant 3T3-L1 adipocytes cultured in galactose, to force ATP-production via mitochondria (*Aguer et al., 2011*), we observed no defect in basal or maximal (uncoupler-induced) respiration (*Figure 7A*). Instead, basal oxygen consumption was increased in multiple models (*Figure 7A*). To explore this further, we measured oxygen consumption in digitonin-permeabilised 3T3-L1 adipocytes to assess maximal respiratory function (*Figure 7B–D*, *Figure 7—figure supplement 1A*). In these experiments, FAD-linked respiratory capacity assessed via succinate was the only activity compromised in all models (*Figure 7C*). This defect was specific to succinate dehydrogenase, since oxidation of medium-chain fatty acid (i.e., octanoylcarnitine, which also donates electrons to CoQ via electron-transferring-flavoprotein dehydrogenase [*Ruzicka and Beinert, 1977*]), was not altered (*Figure 7—figure supplement 1A*). Collectively, these data suggest that loss of mitochondrial CoQ decreases succinate-driven complex II capacity, but this does not limit overall mitochondrial respiration in cells.

Although mitochondrial superoxide (*Hoehn et al., 2009*) and hydrogen peroxide ($H_2O_2$) (*Anderson et al., 2009*; *Paglialunga et al., 2015*) have been implicated in insulin resistance, the cause of increased oxidants has yet to be determined. Elevated mitochondrial $H_2O_2$ in response to lower CoQ content is likely caused by increased production of the superoxide anion radical, the precursor of $H_2O_2$. To test this directly, we determined superoxide in specific in vitro models of adipocyte insulin resistance, using LC-MS to quantify the conversion of mito-hydroethidine to the superoxide-specific product mito-2-hydroxyethidium (*Zielonka et al., 2008*). Mitochondrial superoxide was increased in the CI model (*Figure 7E*) and in adipocytes where CoQ biosynthesis was inhibited (*Figure 7E–7F*), implying that increased mitochondrial $H_2O_2$ following loss of CoQ was due to increased superoxide production.

We next used a series of mitochondrial poisons to determine the site of oxidant production. First, we assessed whether increased $H_2O_2$ was produced by the respiratory chain by incubating cells with chemical uncouplers to depolarise mitochondria (*Fisher-Wellman et al., 2013*). BAM15 and FCCP had no effect on PRDX3 dimerisation in control cells but lowered the PRDX3 dimer/monomer status to control levels in cells treated with NB (*Figure 7G*, *Figure 7—figure supplement 1C*), or in cells in which the expression of *Coq7* and *Coq9* were reduced using siRNA (*Figure 7—figure supplement 1D–E*). This established that loss of CoQ increased $H_2O_2$ in a coupled respiration-dependent manner. Administration of the complex II inhibitors TTFA (*Figure 7—figure supplement 1B*) and malonate lowered the PRDX3 dimer/monomer ratio to near control values in cells treated to inhibit CoQ biosynthesis (*Figure 7G*, *Figure 7—figure supplement 1C–E*) and in other models of insulin resistance (*Figure 7H*, *Figure 7—figure supplement 1H–I*), suggesting that increased $H_2O_2$ in insulin-resistant adipocytes was dependent on complex II. The majority of superoxide from succinate-driven respiration via complex II has been reported to result from reverse electron transport from $CoQH_2$ into complex I, which can be inhibited with rotenone (*Quinlan et al., 2012*). To test whether this may account for $H_2O_2$ generated in response to loss of CoQ we tested whether NB-responsive PRDX3 dimerisation was inhibited by rotenone. Rotenone had no effect on the PRDX3 redox state in NB-treated cells, similar to what was observed for antimycin A (a complex III inhibitor) and oligomycin (a complex V inhibitor) (*Figure 7—figure supplement 1F*). Further, treatment of 3T3-L1 adipocytes with NB did not change the CoQ redox state, just as the CoQ redox state was not altered in other in vitro models of insulin resistance (*Figure 7—figure supplement 1G*). Since a more reduced CoQ pool is required for reverse election transport from $CoQH_2$ to complex I (*Murphy, 2009*), these data suggest that reverse electron transport was not involved in the observed increase in

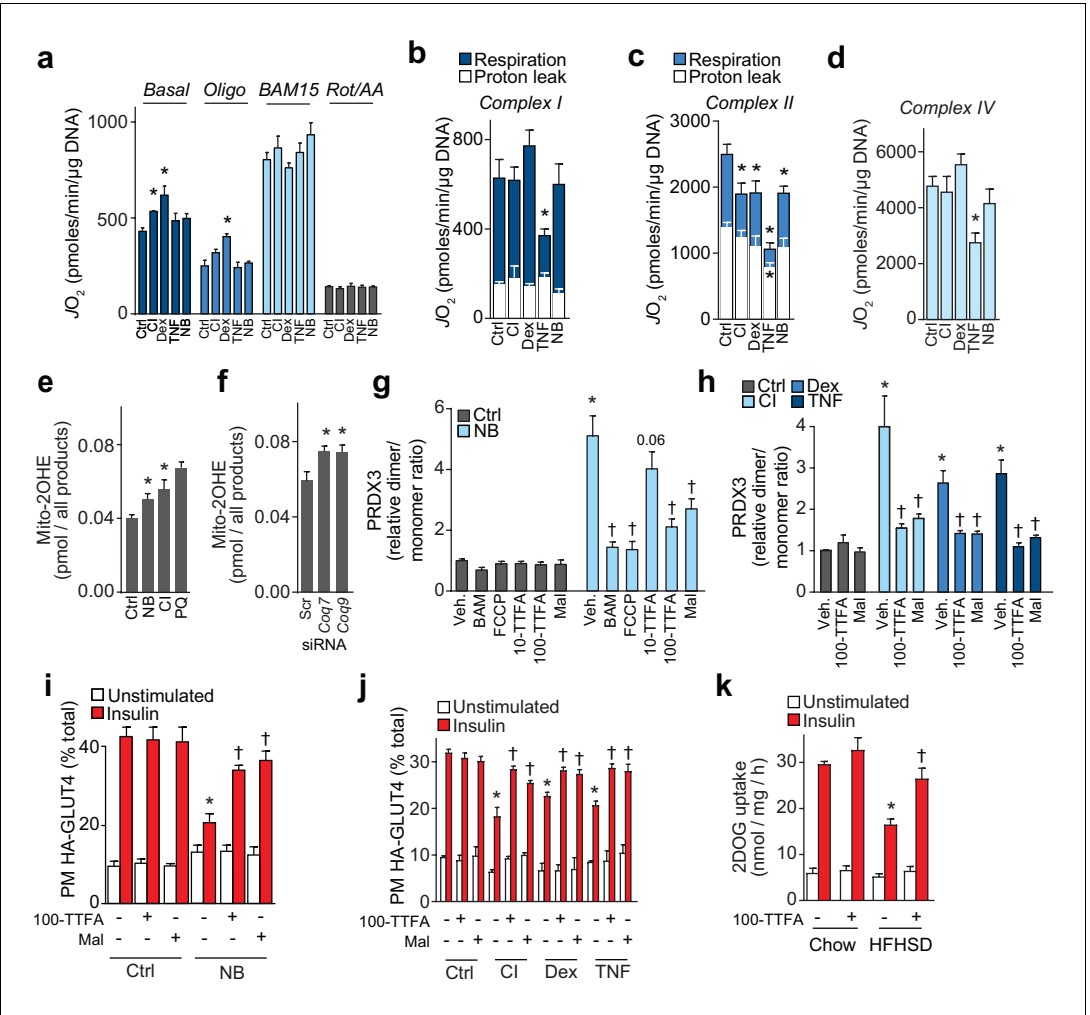

**Figure 7.** Loss of CoQ causes insulin resistance via complex II-derived superoxide/hydrogen peroxide. (A) 3T3-L1 adipocytes were incubated in galactose and respiration was assessed using the XFp Analyzer. Following basal measurements, cells were sequentially treated with, oligomycin (Oligo), BAM15 (uncoupler), and rotenone/ antimycin A (Rot/AA). (B–D) Mitochondrial respiration in 3T3-L1 adipocytes treated to induce insulin resistance as indicated. Cells were permeabilised with digitonin and specific substrates and inhibitors were added to assess complex I (B), II (C) and IV (D) activity as described in the Material and Methods. Results show mean ±S.E.M. of five to eight separate in vitro experiments. *p<0.05 versus control, t-tests corrected for multiple comparisons. (E–F) Concentration of 2-hydroxyethidium derivative of MitoSox in cells treated (Ctrl, control; NB, nitrobenzoic acid; CI; chronic insulin; PQ; paraquat) as indicated and incubated with MitoSox for 1 hr. Results show mean ±S.E.M. for three to eight separate in vitro experiments. *p<0.05 versus control cells, t-tests corrected for multiple comparisons. (G) Relative PRDX3 dimer/monomer ratio in adipocytes treated with or without NB and uncouplers (BAM15, FCCP) or inhibitors of complex II (10 or 100 µM TTFA (10-TTFA; 100-TTFA) and malonate (Mal). Results show mean ±S.E.M. of PRDX3 dimer/monomer ratio relative to control 3T3-L1 adipocytes across four separate in vitro experiments. *p<0.05 versus control cells, †p<0.05 versus NB-treated cells without additional treatment, t-tests corrected for multiple comparisons. (H) Relative PRDX3 dimer/monomer ratio in control or insulin-resistant adipocytes treated with or without inhibitors of complex II (100 µM TTFA or malonate). Results show mean ±S.E.M. of PRDX3 dimer/monomer ratio relative to control 3T3-L1 adipocytes across three separate in vitro experiments. *p<0.05 versus control cells, †p<0.05 versus insulin-resistant cells without additional treatment, t-tests corrected for multiple comparisons. (I) 3T3-L1 adipocytes were treated with NB and TTFA or malonate where indicated before sensitivity was determined by HA-GLUT4 abundance at the plasma membrane (PM). Results show mean ±S. E.M. for four separate in vitro experiments. *p<0.05 versus insulin-stimulated control cells, †p<0.05 versus insulin-stimulated NB-treated cells without additional treatment, t-tests corrected for multiple comparisons. (J) 3T3-L1 adipocytes were treated to induce insulin resistance and with 100 µM TTFA or 10 mM malonate where indicated before sensitivity was determined by HA-GLUT4 abundance at the plasma membrane (PM). Results show mean ±S.

*Figure 7 continued on next page*

*Figure 7 continued*

E.M. for three to four separate in vitro experiments. *$p < 0.05$ versus insulin-stimulated control cells, †$p < 0.05$ versus insulin-stimulated insulin-resistant cells without additional treatment, t-tests corrected for multiple comparisons. (K) Adipose tissue explants from mice fed a chow diet or a HFHSD for 14 d were incubated with or without TTFA prior to assessment of insulin-regulated 2-DOG uptake with 10 nM insulin. Results show mean ±S.E.M. of data from five mice. *$p < 0.05$ versus insulin-stimulated explants from mice fed a chow diet, and †$p < 0.05$ versus insulin-stimulated explants from HFHSD-fed mice without TTFA treatment, t-tests corrected for multiple comparisons. See also *Figure 7—figure supplement 1*.

DOI: https://doi.org/10.7554/eLife.32111.016

The following figure supplement is available for figure 7:

**Figure supplement 1.** Additional data on the effect of loss of CoQ on oxidant production from mitochondrial complex II.

DOI: https://doi.org/10.7554/eLife.32111.017

superoxide/$H_2O_2$ resulting from loss of mitochondrial CoQ, and that complex II itself was the likely origin the oxidants (*Quinlan et al., 2012*).

To test whether inhibition of oxidant production from complex II using TTFA and malonate could overcome insulin resistance we assessed insulin-stimulated HA-GLUT4 translocation in 3T3-L1 insulin-resistant adipocytes in the presence or absence of these complex II inhibitors. Treatment of control 3T3-L1 adipocytes with TTFA did not impair insulin-stimulated HA-GLUT4 translocation (*Figure 7I*) or insulin-regulated inhibition of lipolysis (data not shown), suggesting that impaired complex II activity does not cause adipocyte insulin resistance per se. Both TTFA and malonate improved insulin-stimulated HA-GLUT4 translocation to the PM in all models of insulin resistance tested (*Figure 7I–7J*), albeit not to the same level as observed in control cells. Further, TTFA improved insulin-stimulated 2DOG uptake in adipose explants isolated from mice fed a HFHSD (*Figure 7K*). Taken together, these data suggest that lower mitochondrial CoQ accelerates superoxide generation, most likely from complex II, which in turn elevates the mitochondrial $H_2O_2$ burden, promotes PRDX3 dimerisation and drives insulin resistance.

## Discussion

Mitochondrial oxidants have been reported to play an important role in the development of insulin resistance in adipose (*Hoehn et al., 2009*; *Houstis et al., 2006*; *Paglialunga et al., 2015*) and muscle tissue (*Anderson et al., 2009*; *Hoehn et al., 2009*). This is thought to occur primarily via increased production of superoxide or $H_2O_2$ in mitochondria (*Anderson et al., 2009*; *Hoehn et al., 2009*; *Paglialunga et al., 2015*), yet the proximal mechanism that triggers mitochondrial oxidant production has remained elusive. Here, we provide insights into the sequence of events that lead to increased oxidant production and insulin resistance. Mass-spectrometry-based proteomic analysis of adipose tissue from mice fed a HFHSD, insulin-resistant 3T3-L1 adipocytes and adipose tissue from insulin-resistant humans, revealed down-regulation of the mevalonate/CoQ biosynthesis pathway in multiple models, and a separate targeted metabolite analysis revealed a common decrease in mitochondrial CoQ content in these models as well as in insulin-resistant muscle. Decreased mitochondrial CoQ was sufficient to cause insulin resistance via a mechanism requiring mitochondrial oxidants, while restoration of mitochondrial CoQ restored insulin sensitivity. These data suggest a novel pathway that may drive insulin resistance across a broad range of models including muscle and adipose tissue from mice and adipose tissue from obese humans. The pathway involves decreased expression of mevalonate pathway/CoQ biosynthetic enzymes, lower mitochondrial CoQ and insulin resistance as a result of oxidant production primarily from complex II.

Based on the current study, many insults implicated in insulin resistance including hyperinsulinaemia, inflammation, corticosteroids and caloric excess converge upon loss of mitochondrial CoQ as a potential cause of insulin resistance. Mechanistically, this could be explained by decreased expression of CoQ biosynthetic enzymes, and lower CoQ synthesis rates, in a majority of models studied. Related to this, our finding that mevalonate pathway inhibiting statins lowered CoQ content and caused insulin resistance in a CoQ-dependent manner may shed new light on the link between statin therapy in humans and insulin resistance (*Cederberg et al., 2015*; *Preiss et al., 2011*; *Sattar et al.,*

*2010*). The lack of concordance between transcript and protein expression within the mevalonate/CoQ pathway (*Figure 2—figure supplement 1*) across the different insulin-resistant models suggests that there may be multiple mechanisms by which this pathway is targeted in response to different upstream insults and this maybe mediated via either transcriptional or post-translational regulation. Furthermore, our subcellular analysis of CoQ content and data from the chronic insulin model indicate that other aspects of CoQ biology that we do not yet understand may be involved in regulating mitochondrial CoQ abundance in insulin-resistant conditions. These features could include the regulation of CoQ turnover and/or its trafficking between mitochondria and other parts of the cell. Together, this supports the notion that various insults act in different ways, all decreasing mitochondrial CoQ as a common means of inducing insulin resistance.

Modest loss of CoQ has been reported to increase mitochondrial oxidants in a range of cellular systems (*Cornelius et al., 2013*; *Duberley et al., 2013*; *Quinzii et al., 2013*; *Quinzii et al., 2012*; *Rodríguez-Hernández et al., 2009*), although the precise mechanism for this effect remains unknown. Our data are consistent with loss of mitochondrial CoQ increasing mitochondrial superoxide/$H_2O_2$ production because we observed increased superoxide in response to reduced CoQ biosynthesis using a highly specific mass-spectrometry-based assay for superoxide (*Figure 7E–F*). Our data also suggest that this superoxide is, in part, derived from the flavin site of complex II ($II_F$) (*Quinlan et al., 2012*) since the complex II inhibitors TTFA and malonate lowered PRDX3 dimer/monomer status. Oxidation of succinate via complex II has been reported to generate large amounts of superoxide via reverse electron transport to complex I. However, rotenone had no effect on PRDX3 dimerisation in adipocytes treated with NB and we detected no difference in the overall CoQ redox state, suggesting that under these conditions superoxide/$H_2O_2$ does not originate from complex I via reverse electron transfer. Unlike electron transfer from complex I to CoQ, electron transfer from flavoproteins in complex II to CoQ is not limited by the energetic constraints established by the membrane potential. This means that electrons can be transferred to CoQ whenever additional substrate is made available to these flavoproteins, provided oxidized CoQ is available to receive the electrons. In addition, although there is some evidence that complex III is a site of superoxide production (*Quinlan et al., 2013*), the majority of superoxide in the mitochondria is derived from flavin sites, including complex II (*Quinlan et al., 2013*; *Starkov and Fiskum, 2003*; *Tretter et al., 2007*). Therefore, we hypothesise that increased superoxide production from the $II_F$ site is due to increased steady-state concentrations of the flavin radical as a result of impaired electron transfer to CoQ at the binding site $II_Q$, due to decreased CoQ. This interpretation is supported by our finding that maximal complex II activity was impaired in all models studied. Alternatively, lower CoQ may favour reverse electron transfer to complex II, and superoxide production from $II_F$, under conditions where other enzymes (e.g. mitochondrial glycerol-3-phosphate dehydrogenase [*Orr et al., 2012*]) are feeding electrons into the CoQ pool (*Quinlan et al., 2012*). However, there are likely additional sites of superoxide production since inhibition of complex II only partially rescued PRDX3 dimerisation. Despite knowledge of increased oxidants in insulin-resistant humans (*Boden et al., 2015*) and that scavenging mitochondrial oxidants benefits insulin sensitivity (*Anderson et al., 2009*; *Hoehn et al., 2009*), the mechanism for increased oxidants in mitochondria in insulin resistance has remained unclear. Our data address this question and place loss of mitochondrial CoQ as a common defect and cause of mitochondrial oxidants, via complex II-derived superoxide, and insulin resistance in adipocytes and perhaps muscle.

An important question is how increased mitochondrial oxidants impair insulin action. Our data from cells supplemented with CoQ revealed that improvements in insulin-stimulated glucose transport and inhibition of lipolysis were not associated with improved insulin signalling to Akt or its downstream substrate TBC1D4. This is consistent with previous reports that defects in insulin-stimulated glucose transport in insulin resistance are not due to obvious defects in proximal insulin signalling (*Hoehn et al., 2008*; *Tan et al., 2015*). Although retrograde signalling from the mitochondria to the nucleus is well described it seems unlikely that this mitochondrial oxidant-induced insulin resistance requires changes in transcription because induction of mitochondrial oxidants acutely impairs insulin action (*Hoehn et al., 2009*). Thus, it is more likely that there is a presently undiscovered signal transduction pathway that communicates directly to mediators of insulin action in the cytoplasm. Future studies exploring specific targets involved in this pathway and their connection with mitochondria and oxidants are warranted.

Coenzyme Q has received considerable attention as a supplement to ameliorate a range of medical conditions, including diabetes and cardiovascular disease, based on the observation that serum and tissue $CoQ_{10}$ concentrations are decreased in individuals with these conditions. While there are reports of CoQ supplementation benefitting these conditions (*Amin et al., 2014*; *Ayer et al., 2015*; *Hodgson et al., 2002*; *Mortensen et al., 2014*; *Raygan et al., 2016*), the efficacy of $CoQ_{10}$ in the treatment of diabetes and cardiovascular diseases remains unclear (*Ayer et al., 2015*; *Eriksson et al., 1999*; *Suksomboon et al., 2015*). Our study provides a reasonable rationale for targeting the CoQ biosynthesis pathway as a potential therapeutic target. Overall low bioavailability of orally administered $CoQ_{10}$ or $CoQ_{10}H_2$ represents a substantial limitation, particularly in situations of modest CoQ deficiency such as those shown here to be sufficient to initiate insulin resistance, and where mitochondrial CoQ homeostasis needs to be restored in metabolic tissues such as adipose and muscle (*Zhang et al., 1995*). We overcame this limitation in mice by intra-peritoneal administration of CoQ to provide proof-of-principle that restoration of mitochondrial CoQ improves insulin action and whole body glucose tolerance (*Figure 4*). Unfortunately, intra-peritoneal administration of $CoQ_{10}$ is not likely a practical strategy for the treatment of insulin resistance in humans. Pharmacological inhibition of the CoQ biosynthesis pathway, e.g., by polyisoprenoid epoxides (*Bentinger et al., 2008*), or targeting additional processes that contribute to the regulation of mitochondrial CoQ content may represent potential options in the future. In the present study, we found that the protein levels, but not the corresponding levels of the mRNAs, of the CoQ biosynthetic enzymes COQ7 and COQ9 were decreased in insulin resistance. These proteins form a dimeric complex (*Lohman et al., 2014*) the formation of which may be regulated via a post-translational mechanism. Although little is known about the molecular regulation of COQ protein turnover, the CoQ biosynthetic protein complex has been reported to be stabilised by the atypical kinase ADCK3/COQ8 (*He et al., 2014*; *Stefely et al., 2016*)), mitochondrial proteases (*Veling et al., 2017*) and CoQ itself (*He et al., 2014*). Destabilisation of the complex lowers expression of many of the COQ proteins altered in insulin-resistant adipose tissue, including COQ3, 7 and 9. Therefore, it may be that loss of COQ7 and 9 in insulin resistance is a result of increased turnover due to complex instability. Another therapeutic option is based on our data implicating complex II as the site of increased oxidant production in response to loss of mitochondrial CoQ. Recent chemical screens have successfully identified compounds that prevent superoxide production from complex I (*Brand et al., 2016*) and III (*Orr et al., 2015*), without impairing electron transport. Identifying similar compounds for complex II may be useful in mitigating superoxide/$H_2O_2$ production and overcoming insulin resistance.

# Materials and methods

**Key resources table**

| Reagent type (species) or resource | Designation | Source or reference | Identifiers |
| --- | --- | --- | --- |
| Strain, strain background (mouse) | C57/Bl6J mice | Animal Resources Centre (Perth, Australia) or Australian BioResources (Moss Vale, Australia) | RRID:IMSR_JAX:000664 |
| Cell line (mouse) | Mouse: 3T3-L1 adipocytes | Dr Howard Green, Harvard Medical School | RRID:CVCL_0A20 |
| Biological sample (human) | Human adipose tissue | Clinical details in PMID: 26378474 | |
| Antibody | Mitochondrial complex subunits (OXPHOS) | Thermo Fisher Scientific | Cat# 45–8099 |
| Antibody | PRDX1 | Thermo Fisher Scientific | Cat# PA3-750 |
| Antibody | PRDX2 | Abcam | Cat# ab109367, clone: EPR5154 |
| Antibody | catalase | Abcam | Cat# ab52477 |
| Antibody | PRDX3 | Ab Frontier | Cat# LF-PA0030 |
| Antibody | 14-3-3 | Santa Cruz | Cat# sc-629, clone K19 |

*Continued on next page*

Continued

| Reagent type (species) or resource | Designation | Source or reference | Identifiers |
|---|---|---|---|
| Antibody | pT642 TBC1D4 | Cell Signaling Technologies | Cat# 4288 |
| Antibody | TBC1D4 | Cell Signaling Technologies | Cat# 2670 |
| Antibody | pT308 Akt | Cell Signaling Technologies | Cat# 9275 |
| Antibody | pS473 Akt | Cell Signaling Technologies | Cat# 4051 |
| Antibody | Akt | Cell Signaling Technologies | Cat# 4685 |
| Antibody | $\alpha$-Tubulin | Sigma Aldrich | Cat# T9026 |
| Antibody | GLUT4 | In-house | NA |
| Antibody | anti-HA antibody | Covance | clone 16B1 |
| Sequence-based reagent | qPCR primers: mCoq7_F; tttggaccatagctgcattg and mCoq7_R; tgaggcctcttccatactctg, | Sigma Aldrich | NA |
| Sequence-based reagent | qPCR primers: mCoq9_F; tcagcagcattctgagacaca and mCoq9_R; gtgctgtagctgctcctcact, | Sigma Aldrich | NA |
| Sequence-based reagent | qPCR primers: mCypB-F; ttcttcataaccacagtcaagacc; mCypB-R, accttccgtaccacatccat. | Sigma Aldrich | NA |
| Sequence-based reagent | Scrambled siRNA: (sense 5'-CAGTCGCGTTTGCGACTGGTT-3') | Shanghai Genepharma | NA |
| Sequence-based reagent | anti-Coq7 siRNA (sense 5'-GGGAUCACGCUGGUGAAUAUTT-3', 5'-GGAUGACCUUAGACAAUAUTT-3', 5'-GCCUUGUUGAAGAGGAUUAUTT-3') | Shanghai Genepharma | NA |
| Sequence-based reagent | anti-Coq9 siRNA (sense 5'-GCAGCAUUCUGAGACACAGTT-3', 5'-GCUGGUGAUGAUGCAGGAUTT-3', 5'-GCAAUGAACAUGGGCCACATT-3') | Shanghai Genepharma | NA |
| Commercial assay or kit | Glycerol assay kit | Sigma Aldrich | Cat#: FG0100-1KT |
| Commercial assay or kit | NEFA kit | Waki Pure Chemical Industries | Cat#: 279–75104 |
| Commercial assay or kit | Insulin ELISA | Crystal Chem | Cat#: 90080 |
| Commercial assay or kit | RNeasy kit | QIAGEN | Cat#: 74104 |
| Commercial assay or kit | PrimeScript first strand cDNA synthesis kit | Clontech | Cat#: 6110A |
| Chemical compound, drug | 2-[1,2-3H(N)]-deoxy-D-glucose | Perkin Elmer | Cat# NET328001MC |
| Chemical compound, drug | Insulin from bovine pancreas | Sigma-Aldrich | Cat# I5500-1G; CAS 11070-73-8 |
| Chemical compound, drug | Coenzyme $Q_9$ | Sigma Aldrich | Cat# 27597–1 MG CAS 303-97-9 |
| Chemical compound, drug | $CoQ_{10}$ liposomal formulation | Tichson Corp | LiQsorb |
| Chemical compound, drug | 4-Chlorobenzoic acid | Sigma Aldrich | Cat# 135585–50G CAS 74-11-3 |
| Chemical compound, drug | 4-Nitrobenzoic acid | Sigma Aldrich | Cat# 461091–250G CAS 62-23-7 |
| Chemical compound, drug | NEM | Sigma Aldrich | Cat# 04259–5G CAS 128-53-0 |
| Chemical compound, drug | FCCP | Sigma Aldrich | Cat# C2920-10MG CAS 370-86-5 |

*Continued*

| Reagent type (species) or resource | Designation | Source or reference | Identifiers |
|---|---|---|---|
| Chemical compound, drug | BAM15 | Dr Kyle Hoehn | PMID: 24634817 |
| Chemical compound, drug | TTFA | Sigma Aldrich | Cat# T27006-25G CAS 326-91-0 |
| Chemical compound, drug | Malonate | Sigma Aldrich | Cat# M1296-100G CAS 141-82-2 |
| Chemical compound, drug | Isoproterenol | Sigma Aldrich | Cat# I6504-100MG CAS 5984-95-2 |
| Chemical compound, drug | MitoSOX | Thermo Fisher Scientific | Cat# M36008 |
| Software, algorithm | MaxQaunt | https://www.biochem.mpg.de/5111795/maxquant | Versions 1.5.7.0, 1.4.0.8, 1.3.0.5 |
| Software, algorithm | Affeymetrix GeneChip Command Console Software | Thermo Fisher Scientific | NA |
| Software, algorithm | R-programming environment – affy package | http://bioconductor.org/packages/affy/ | Version 1.52.0 |
| Software, algorithm | R-programming environment – limma package | http://bioconductor.org/packages/limma/ | Version 3.30.3 |
| Software, algorithm | R-programming environment – directPA package | https://CRAN.R-project.org/package=directPA | Version 1.3 |
| Software, algorithm | R-programming environment – Re-Fraction package | https://github.com/PengyiYang/Re-Fraction (copy archived at https://github.com/elifesciences-publications/Re-Fraction) | Version 1.2 |
| Software, algorithm | Image lab 5.2.1 | BioRad | NA |
| Software, algorithm | Image Studio | LiCOR | NA |
| Software, algorithm | Graphpad Prism | GraphPad Software Inc. | NA |
| Deposited Data | Mouse and cell line mass spectrometry data. | ProteomeXchange Consortium | PXD005128. |
| Deposited Data | Human mass spectrometry data. | ProteomeXchange Consortium | PXD006891 |
| Deposited Data | Mouse and cell line microarray data. | https://www.ncbi.nlm.nih.gov/geo/query/acc.cgi?acc=GSE87853 | GSE87853 |
| Deposited Data | Mouse and cell line microarray data. | https://www.ncbi.nlm.nih.gov/geo/query/acc.cgi?acc=GSE87854 | GSE87854 |

## Animal details

Eight-week-old male C57BL/6J mice were purchased from the Animal Resources Centre (Perth, Australia) or Australian BioResources (Moss Vale, Australia). The animals were kept in a temperature-controlled environment (22 ± 1°C) on a 12 hr light/dark cycle with free access to food and water. Mice were fed *ad libitum* for a period of 14 days with a standard lab diet (CHOW) (13% calories from fat, 22% calories from protein, and 65% calories from carbohydrate, 3.1 kcal/g; Gordon's Specialty Stock Feeds, Yanderra, Australia) or with high fat high sucrose diet (HFHSD; 47% of calories from fat (40% calories from lard), 21% calories from protein, and 32% calories from carbohydrates (16% calories from starch), 4.7 kcal/g). All experiments were carried out with the approval of the Garvan Institute/St. Vincent's Hospital Animal Experimentation Ethics Committee (09/46) or the approval of the University of Sydney Animal Ethics Committee (2014/694), following guidelines issued by the National Health and Medical Research Council of Australia. All studies used at least five mice per treatment group. Mice from different cages were used to negate cage-specific effects.

## 2DOG uptake assays in adipose tissue explants

Epididymal fat depots were excised from mice, transferred immediately to warm DMEM/2% BSA/20 mM HEPES, pH 7.4, and minced into fine pieces. Explants were washed twice and incubated in

DMEM/2% BSA/20 mM HEPES, pH 7.4 for 2 hr. Explants were then rinsed and incubated in Krebs–Ringer phosphate buffer containing 2% bovine serum albumin (BSA, Bovostar, Bovogen) (KRP buffer; 0.6 mM $Na_2HPO_4$, 0.4 mM $NaH_2PO_4$, 120 mM NaCl, 6 mM KCl, 1 mM $CaCl_2$, 1.2 mM $MgSO_4$ and 12.5 mM Hepes (pH 7.4)). Insulin was added for 20 min, and glucose transport was initiated by addition of 2-DOG (0.25 µCi, 50 µM) and [$^{14}$C]mannitol (Source, 0.036 µCi/sample) for the final 5 min of the assay to measure steady-state rates of 2DOG uptake. For experiments using TTFA (*Figure 7K*), 100 µM TTFA or equivalent volume of vehicle (EtOH) was included during the 2 hr incubation period in DMEM/2% BSA and maintained throughout subsequent washes and incubation. Also, the 2DOG uptake assay was carried out in DMEM without glucose/2% BSA rather than KRP/2% BSA. Uptake was terminated with three rapid washes in ice-cold PBS, after which the cells were solubilised in radioimmune precipitation assay buffer (RIPA; 50 mM Tris-HCl, pH 7.5, 150 mM NaCl, 1% Triton X-100, 0.5% sodium deoxycholate, 0.1% SDS, 1 mM EDTA, and 10% glycerol) supplemented with protease inhibitors (Roche). Samples were assessed for radioactivity by scintillation counting and the results were normalised for protein content determined by the bicichoninic acid assay. Cages were randomly assigned to diets and investigators were not blinded to experimental groups.

## In vivo metabolic assays

Glucose tolerance tests (GTTs) and insulin tolerance tests (ITTs) were performed on mice following a 6 hr fast from 0700 to 1300. For GTTs, mice were injected i.p. with 10% glucose solution at 1 g/kg (*Figure 1*) or 2 mg/kg (*Figure 4* and *Figure 4—figure supplement 1*) per lean mass. For tracer uptake during the GTT, mice were also administered [$^3$H]- 2DOG tracer (200 µCi/kg lean weight) within the 10% glucose solution. For ITTs, mice were i.p. injected with 80 mg/kg pentobarbitone sodium (Lethabarb Euthanasia Injection, Virbac, Australia), after 20 min the abdominal cavity was incised along the midline to reveal the liver to inject 1 U/kg lean weight of insulin and [$^3$H]- 2-DOG (200 µCi/kg lean weight) into the hepatic portal vein. At the times indicated, blood was sampled from the tail tip and blood glucose determined with an Accu-Chek II glucometer (Roche). Clearance of the [$^3$H]- 2DOG tracer from the blood during the GTT and ITT was assessed to allow calculation of tracer disappearance. [$^3$H]- 2DOG tracer uptake into epididymal and inguinal adipose tissue and quadriceps muscle and conversion to glucose-6-phospate was determined as previously described (*Smith et al., 2007*). Plasma NEFAs were measured using NEFA C (Wako, Osaka, Japan) according to the manufacturer's instructions. Cages were randomly assigned to diets and investigators were not blinded to experimental groups.

## Assessment of blood insulin during GTT

Blood samples were obtained via tail bleeds using 5 µL heparinised hematocrit tubes (Drummond) and ejected into a mouse ultra-sensitive insulin ELISA (90080, Crystal Chem). ELISA performed as per manufacturer's instructions.

## 3T3-L1 fibroblast culture and differentiation into adipocytes

Mycoplasma-free 3T3-L1 fibroblasts obtained from 3T3-L1 Howard Green (Harvard Medical School, Boston, MA) were maintained in Dulbecco's Modified Eagle Medium (DMEM) (Thermo Fisher Scientific) supplemented with 10% fetal calf serum (FCS) (Thermo Fisher Scientific), 1% GlutaMAX (Thermo Fisher Scientific) in a humidified atmosphere with 10% $CO_2$. HA-GLUT4 overexpressing 3T3-L1 fibroblasts were generated by retroviral transduction as previously described (*Govers et al., 2004*). Confluent 3T3-L1 cells were differentiated into adipocytes by the addition of DMEM containing 0.22 µM dexamethasone, 100 ng/mL biotin, 2 µg/mL insulin, 500 µM IBMX (day 0). After 72 hr, medium was replaced with DMEM/10% FCS/GlutaMAX containing 2 µg/mL insulin (day three post differentiation). After a further 72 hr (day six post differentiation), cells were cultured in DMEM/10% FCS/GlutaMAX. Medium was subsequently replaced every 48 hr. Cells were used between days 10 and 15 after the initiation of differentiation.

For stable isotope labelling of amino acids in cell culture (SILAC)-based proteomics, 3T3-L1 fibroblasts were passaged for six cell divisions in DMEM (Sigma Alrich)/10% dialysed FCS (Thermo Fisher Scientific) containing L-arginine (Arg 0) and L-lysine (Lys 0) ('light'), L-arginine-U-$^{13}$C$_6$$^{14}$N$_4$ (Arg 6) and L-lysine-$^2$H$_4$ (Lys 4) ('medium') or L-arginine-U-$^{13}$C$_6$$^{15}$N$_4$ (Arg 10) and L-lysine-U-$^{13}$C$_6$$^{15}$N$_2$ (Lys 8) ('heavy'). Final concentrations of arginine and lysine were 33 µg/mL and 76 µg/mL, respectively. This

strategy generated three distinct SILAC populations. We periodically tested labelling efficiency by mass spectrometry analysis. SILAC-labelled fibroblasts were differentiated into adipocytes as above.

## In vitro models of insulin resistance and CoQ treatment

Insulin resistance was induced by dexamethasone, tumour necrosis factor-α (TNF), or hyperinsulinaemia as previously described (*Hoehn et al., 2008*). The chronic insulin (CI) model of hyperinsulinemia was created by addition of 10 nM insulin to adipocytes at 1200, 1600 and 2000 hr on day 1 and 0800 hr the following day. Glucocorticoid-induced insulin resistance was recreated with 20 nM dexamethasone (Dex) (0.01% ethanol carrier as control), starting on day seven post initiation of differentiation and media was changed every other day for 8 d. Chronic low-dose inflammation was mimicked in 3T3-L1 adipocytes by incubation with 2 ng/mL TNFα (Calbiochem) for 4 d. Medium was changed every 24 hr. For CoQ treatment, cells were incubated with 10 μM CoQ$_9$ (Sigma Aldrich) or 10 μM liposomal CoQ$_{10}$ (LiQsorb, Tichson Corp.) for 24 hr prior to assays. CoQ$_9$ was dissolved in ethanol at 5 mM and diluted to 10 μM in pre-warmed DMEM/10% FCS, 1% GlutaMAX and incubated at 37°C for 30 min prior to addition to cells. Ethanol was used as a vehicle control (0.2%).

## 2-Deoxyglucose uptake assays in cultured cells

Following 2 hr serum-starvation in DMEM/0.2% BSA/1% GlutaMAX, cells were washed and incubated in pre-warmed Krebs–Ringer phosphate buffer containing 0.2% bovine serum albumin (BSA, Bovostar, Bovogen) (KRP buffer; 0.6 mM Na$_2$HPO$_4$, 0.4 mM NaH$_2$PO$_4$, 120 mM NaCl, 6 mM KCl, 1 mM CaCl$_2$, 1.2 mM MgSO$_4$ and 12.5 mM Hepes (pH 7.4)). Cells were stimulated with 100 nM insulin for 20 min. To determine non-specific glucose uptake, 25 μM cytochalasin B (ethanol, Sigma Aldrich) was added to the wells before addition of 2-[$^3$H]deoxyglucose (2-DOG) (PerkinElmer). During the final 5 min 2-DOG (0.25 μCi, 50 μM) was added to cells to measure steady-state rates of 2DOG uptake. Following three washes with ice-cold PBS, cells were solubilised in PBS containing 1% (v/v) Triton X-100. Tracer uptake was quantified by liquid scintillation counting and data normalised for protein content. Data were further normalised to maximal insulin stimulation of control cells, set to 100%.

## HA-GLUT4 assay

Determination of plasma membrane HA-GLUT4 was performed as previously described (*Govers et al., 2004*). Briefly, cells were serum-starved for 2 hr in DMEM/0.2% BSA/GlutaMAX. Cells were stimulated with 100 nM insulin for 20 min as indicated. Cells were fixed but not permeabilised, and the amount of HA-GLUT4 present at the plasma membrane determined by the accessibility of the HA epitope to anti-HA antibody (Covance, clone 16B12). Cells were incubated with 20 μg/mL goat anti-mouse Alexa-488-conjugated secondary antibody (Thermo Fisher Scientific). Determination of total HA-GLUT4 was performed in a separate set of cells that underwent the same labelling procedure except that anti-HA staining was performed after permeabilisation of the cells with 0.1% (w/v) saponin. Total HA-GLUT4 was measured separately for each experimental treatment group. Fluorescence (excitation 485 nm/emission 520 nm) was measured using a fluorescent microtiter plate reader (FLUOstar Galaxy, BMG LABTECH). Surface HA-GLUT4 was expressed as a percentage of total HA-GLUT4.

## Lipolysis assay

Following 2 hr serum-starvation in DMEM/0.2% BSA/1% GlutaMAX, cells were washed and incubated in DMEM containing 3.5% fatty acid-free BSA (Sigma-Aldrich), GlutaMAX and 10 mM glucose. Cells were treated with or without 1 nM isoproterenol and/or indicated doses of insulin for 1 hr. Aliquots of medium were taken to assay for glycerol content using Sigma glycerol reagent (Sigma-Aldrich) according to the manufacturer's protocol. Following three washes with ice-cold PBS, cells were solubilised in PBS containing 1% (v/v) Triton X-100. Glycerol release as a measure for lipolysis was normalized to cellular protein content. Complete datasets are presented in *Figure 4—figure supplement 1* and *Figure 5—figure supplement 1*. Insulin-stimulated inhibition of lipolysis (*Figures 4C*, *5C and F*) was calculated as glycerol release from cells treated with 0.5 nM insulin and 1 nM isoproterenol as a percentage of the glycerol release from cells treated with 1 nM isoproterenol alone.

## Human cohort and protocol

Adipose mitochondrial CoQ content was assessed in a cohort of females studied using 2-stage hyperinsulinaemic-euglycaemic clamps with deuterated glucose tracers and adipose biopsies, as previously described (*Chen et al., 2015*). Briefly, the study protocol was approved by St Vincent's Hospital Human Research Ethics Committee (Sydney, Australia) (HREC/10/SVH/133) and written consent obtained prior to the study. Volunteers were sedentary individuals with obesity (BMI >30 kg/m$^2$) and exclusion criteria were diabetes, treatment with medications known to affect glucose homeostasis, >20 g/d alcohol intake, body weight instability in previous 3 months and known cancer, cardiac, renal or liver disease. All studies were performed at the Clinical Research Facility at the Garvan Institute of Medical Research (Sydney, Australia). The clamp started with a 2 hr primed (5 mg/kg), continuous (3 mg/kg/h) infusion of [6,6-$^2$H$_2$]glucose, followed by a 2 hr infusion of low-dose insulin (15 mU/m$^2$/min) and a 2 hr infusion of high-dose insulin (80 mU/m$^2$/min). The deuterated glucose infusion rate was halved (1.5 mg/kg/h) during, and ceased at the end of, the low-dose insulin infusion. Glucose was infused to maintain whole-blood concentration of 5 mmol/L with variable rate infusion of dextrose (25%, enriched to 2.5% with deuterated glucose). Endogenous glucose production and non-esterified fatty acid (NEFA) suppression during the low- dose insulin clamp reflect hepatic and adipose insulin resistance, while glucose infusion rate during the last 30 min of the high-dose insulin clamp normalised to body fat free mass estimates muscle insulin resistance (*Chen et al., 2015*). Periumbilical subcutaneous fat biopsy was performed during the basal clamp stage under sterile conditions using a trocar, as described (*Chen et al., 2015*). Adipose tissue (50 mg) was fixed, dehydrated, paraffin embedded and sectioned and adipocyte cell size measured as previously described (*Chen et al., 2015*) or snap frozen and stored in −80°C for processing and analysis of CoQ content. Body composition was evaluated using dual-energy X-ray absorptiometry and abdominal fat distribution and liver fat by magnetic resonance imaging as previously described (*Chen et al., 2015*). To separate subjects into groups (e.g. high vs low mitochondrial CoQ$_{10}$ concentrations as in *Table 1*, or insulin sensitive vs insulin resistant in *Figure 3E* and *Figure 3—figure supplement 1K*), subjects were divided into an upper tertile (typically 11 subjects) or lower two tertiles (typically 22 subjects) based on specified parameters (e.g. mitochondrial CoQ$_{10}$ concentration (*Table 1*), suppression of NEFAs (*Figure 3E*), glucose infusion rates [*Figure 3—figure supplement 1K*]).

## Sample preparation for mass spectrometry (mouse adipose tissue)

Epididymal adipose tissue was excised from mice fed a HFHSD for the specified durations (five mice per time point). Tissue was lysed in 6 M urea, 2 M thiourea, 25 mM triethylammonium bicarbonate, pH 7.9 containing phosphatase and protease inhibitor cocktails (Roche) by tip-probe sonication (2 × 30 s) on ice. Lysates were centrifuged at 17,000 x g, 15 min, 4°C. The fat cake was removed and the supernate precipitated with 6 volumes of acetone, overnight at −20°C. Pelleted protein was re-suspended in 6 M urea, 2 M thiourea, 25 mM triethylammonium bicarbonate, pH 7.9 and quantified by Qubit fluorescence (Thermo Fisher Scientific). 100 µg of protein was subjected to reduction with 10 mM DTT for 60 min at 25°C and alkylated with 25 mM iodoacetamide for 30 min at 25°C in the dark. Excess iodoacetamide was then removed by reaction 20 mM DTT and the sample digested with Lys-C (Wako) at 1:50 (w/w?) enzyme to substrate ratio for 2 hr at 25°C. The mixture was diluted 5-fold with 25 mM triethylammonium bicarbonate and digested further with trypsin at 1:50 enzyme to substrate ratio for 12 hr at 30°C. The peptide mixture was acidified to a final concentration of 2% formic acid, 0.1% trifluoroacetic acid and centrifuged at 16,000 x g for 15 min. Peptides were desalted using hydrophilic lipophilic balance – solid phase extraction (HLB-SPE) cartridges (Waters) followed by elution with 50% acetonitrile, 0.1% trifluoroacetic acid and dried by vacuum centrifugation.

## Sample preparation for mass spectrometry (3T3-L1 adipocytes)

SILAC 3T3-L1 adipocytes were left untreated (insulin sensitive) or treated to induce insulin resistance (insulin resistant) as described above. Four biological replicates were used for all SILAC experiments. SILAC mixes were made so that a SILAC doublet contained a constant untreated sample acting as a reference for each insulin-resistant model. A label switch was performed in biological replicates to ensure that identified changes in protein expression were due to experimental manipulation. For example, the light versus heavy doublet was analysed twice; we compared protein content in both

insulin sensitive (light) versus insulin resistant (heavy) and insulin sensitive (heavy) versus insulin resistant (light).

Cells were serum-starved for 2 hr, washed three times with ice-cold PBS and lysed in RIPA containing protease inhibitors. After 15 min on ice, the protein concentration was determined by the BCA assay (Thermo Fisher Scientific). Insulin-resistant samples were mixed in a 1:1 ratio based on total protein concentration with an insulin sensitive control to form a SILAC doublet. Per SILAC doublet, two samples were prepared as follows: mixed lysates were centrifuged at 18,000 x g for 15 min at 4°C to pellet insoluble material, and the resulting supernate (RIPA-soluble fraction) collected. The pellet was re-dissolved in RIPA buffer containing 4% SDS, sonicated (Bandelin SONOPLUS) (1 s × 12), and centrifuged at 18,000 x g for 15 min, and the resulting supernate (RIPA-insoluble fraction) collected. The pH of the RIPA-soluble and –insoluble fraction was adjusted to ~8 by the addition of 50 mM Tris, pH 8.8. Protein thiols were reduced by incubating samples with 1 mM dithiothreitol (DTT) at 95°C for 3 min, followed by incubation for 25 min at room temperature and alkylation with 5.5 mM iodoacetamide in the dark at room temperature for 20 min. Proteins were precipitated in 5 volumes of acetone at −20°C overnight. Precipitated proteins were pelleted at 15,000 x g, allowed to air dry before being re-suspended in 4% SDS, 50 mM Tris, pH 6.8. The protein concentration of each sample was then determined by the BCA assay.

RIPA-soluble and RIPA-insoluble samples for each SILAC doublet were prepared for SDS-PAGE by diluting in sample buffer (Thermo Fisher Scientific). Samples were separated by SDS-PAGE on 4–20% gradient gels (Thermo Fisher Scientific). Gels were stained with SYPRO Ruby Protein Gel Stain (Thermo Fisher Scientific) following the manufacturer's instructions. Lanes containing RIPA-soluble samples were cut into 11 slices, and lanes containing RIPA-insoluble samples were divided into eight slices. Each slice contained approximately the same level of SYPRO Ruby signal and was cut into 1 mm x 1 mm squares to assist protein extraction. Gel pieces were de-stained in 250 mM ammonium bicarbonate pH 8, 50% acetonitrile under constant agitation for 30 min. Destain solution was removed and gel pieces dried in 100% acetonitrile. After 10 min, acetonitrile was removed and proteins were digested by the addition of modified trypsin (Promega, 12.5 ng/μL) in 100 mM NH$_4$HCO$_3$. Digestion was carried out overnight at 37°C. The digestion was stopped by the addition of 5% formic acid and peptides eluted with acetonitrile. Finally, peptide solutions were dried, re-suspended in 1% trifluoroacetic acid and desalted on C18 Stagetips (3M, Empore).

## Sample preparation for mass spectrometry (human adipose tissue)

Adipose tissue homogenates were diluted 1:1 in 6 M guanidine in 100 mM Tris pH 7.5 containing 10 mM Tris(2-carboxyethyl)phosphine and 40 mM chloroacetamide, and heated at 95°C for 5 min. The lysate was tip-probe sonicated and centrifuged at 20,000 x g for 30 min at 4°C. The supernatant was precipitated with 6 volumes of acetone, overnight at −20°C. Pelleted protein were re-suspended in 10% trifluoroethanol in 100 mM Tris, pH 7.5 and quantified by BCA. Seven μg of protein was digested with 140 ng sequencing grade Lys-C (Wako) for 2 hr at 25°C followed by 140 ng of sequencing grade trypsin (Sigma Aldrich) overnight at 37°C. The digest was acidified to a final concentration of 1% trifluroacetic acid (TFA), and peptides purified using SDB-RPS solid-phase disks (Sigma Aldrich) and eluted with 1% ammonium hydroxide in 80% acetonitrile. Peptides were dried by vacuum centrifugation and resuspended in 0.1% TFA in 2% acetonitrile.

## Quantitative mass spectrometric analysis

For mouse adipose tissue experiments, peptides were loaded onto a 50 cm column with 75 μm inner diameter, packed in-house with 1.9 μM C18 ReproSil particles (Dr Maisch GmbH). Reversed-phase chromatography was performed on an Easy nLC1000 HPLC using a binary buffer system of 0.5% acetic acid (buffer A) and 80% acetonitrile/0.5% acetic acid (buffer B). Peptides were separated by linear gradients of buffer B from 5% to 35% over 240 min, at a flow rate of 250 nL/min and electrosprayed into the mass spectrometer by the application of 2.3 kV using a liquid junction connection. Ionised peptides were analysed on Q-Exactive (Thermo Fisher Scientific). The Q-Exactive was operated in data-dependent acquisition mode, acquiring survey scans of 3 million ions at a resolution of 70,000 at 200 m/z. Twenty of the most abundant isotope patterns from each of the survey scans with charge state ≥2 were selected with an isolation window of 2 Th, and fragmented in the HCD cell with NCE of 25. Maximum ion fill times for the MS/MS scans were 120 ms, target fill value was 1e$^5$ ions with an

under fill ratio of 20%. Fragmented ions were analysed with high resolution (35,000 at 400 m/z) in the Orbitrap analyser. Dynamic exclusion was enabled with a duration of 60 s.

For SILAC cell culture experiments, peptides were loaded onto a 20 cm column with 75 μm inner diameter, packed in-house with 3 μM C18 ReproSil particles (Dr Maisch GmbH). Reversed-phase chromatography was performed on either the Dionex Ultimate 3000 or the Easy nLC II HPLC using a binary buffer system of 0.5% acetic acid (buffer A) and 80% acetonitrile/0.5% acetic acid (buffer B). Peptides were separated by linear gradient of buffer B from 5% to 35% over 240 min, at a flow rate of 250 nL/min and electrosprayed into the mass spectrometer by the application of 1.9–2.3 kV using a liquid junction connection. Ionised peptides were analysed on an Orbitrap Velos (Thermo Fisher Scientific).

The Orbitrap Velos was operated in data-dependent acquisition mode, acquiring survey scans of 1 million ions at a resolution of 30,000 at 400 m/z. Three survey-scan ranges (350–1050 m/z, 850–1750 m/z and 350–1750 m/z) were utilised to enhance dynamic range. Five (for the first two mass windows) or seven (for the last mass window) of the most abundant isotope patterns from each of the survey scans with charge state ≥2 were selected with an isolation window of 2 Th, and fragmented in the HCD cell with NCE of 40. Maximum ion fill times for the MS/MS scans were 150 ms, target fill value was $4e^4$ ions, and ion selection thresholds were $1e^4$. Fragmented ions were analysed with high resolution (7500 resolution at 400 m/z) in the Orbitrap analyser. Dynamic exclusion was enabled with a duration of 60 s and a mass window of ±7 ppm. Lock-mass was enabled using 445.120025. On the Q-Exactive, survey scans (300–1600 m/z) were acquired at a resolution of 70,000 and up to 15 of the most intense isotope patterns were selected for fragmentation in the HCD cell with NCE 25. Ion fill targets for MS/MS were 1E5 ions and fragmented ions were with high resolution (17,500) in the Orbitrap analyser.

For human adipose tissue experiments, peptides were loaded onto a 50 cm column with 75 μm inner diameter, packed in-house with 1.9 μM C18 ReproSil particles (Dr Maisch GmbH). Reversed-phase chromatography was performed on an Easy nLC1200 HPLC using a binary buffer system of 0.1% formic acid (buffer A) and 80% acetonitrile/0.5% acetic acid (buffer B). Peptides were separated by linear gradients of buffer B from 2% to 35% over 180 min, at a flow rate of 300 nL/min and electrosprayed into the mass spectrometer by the application of 2.4 kV using a liquid junction connection. Ionised peptides were analysed on Q-Exactive HF (Thermo Fisher Scientific) operated in data-dependent acquisition mode, acquiring survey scans of 3 million ions at a resolution of 60,000 at 200 m/z. Fifteen of the most abundant isotope patterns from each of the survey scans with charge state ≥2 were selected with an isolation window of 1.4 Th, and fragmented in the HCD cell with NCE of 27. Maximum ion fill times for the MS/MS scans were 125 ms, target fill value was $2e^5$ ions with a target threshold set to $1e^5$ Fragmented ions were analysed with high resolution (15,000 at 200 m/z) in the Orbitrap analyser. Dynamic exclusion was enabled with a duration of 60 s.

## Mass spectrometry data processing

For mouse adipose tissue experiments, raw mass spectrometry data were processed using the Max-Quant software (*Cox and Mann, 2008*; *Cox et al., 2009*) version 1.4.0.8 using the default settings with minor changes: Oxidised Methionine (M) and Acetylation (Protein N-term) were selected as variable modifications, and carbamidomethyl (C) as fixed modification. A maximum of two missed cleavages was permitted, 10 peaks per 100 Da, MS/MS tolerance of 20 ppm, and a minimum peptide length of 6. The 'matching between runs' algorithm was enabled with a time window of 2 min to transfer identifications between adjacent samples. Database searching was performed using the Andromeda search engine (*Cox et al., 2011*) integrated into the MaxQuant environment against the mouse Uniprot proteome database, concatenated with known contaminants and reversed sequences of all entries. Protein, peptide and site FDR was controlled at a maximum of 1% respectively. All contaminants and reverse sequenced peptides were removed and the data were log2 transformed. Proteins with 60% or more missing values within each experimental group were removed. Missing expression data within each group was imputed using nearest neighbour averaging or K-nearest neighbours (k = 5) from R package impute (*Trevor Hastie et al., 2018*).

For SILAC cell culture experiments, raw mass spectrometry data were processed using the Max-Quant software (*Cox and Mann, 2008*; *Cox et al., 2009*) version 1.3.0.5 using the default settings with minor changes: oxidised methionine (M) and acetylation (Protein N-term) were selected as variable modifications, and carbamidomethyl (C) as fixed modification, as well as double SILAC labels

(either Arg 0/Lys 0 and Arg 10/Lys 8 or Arg 6/Lys 4 and Arg 10/Lys 8). A maximum of two missed cleavages was permitted, 10 peaks per 100 Da, MS/MS tolerance of 20 ppm, and a minimum peptide length of 6. The 'matching between runs' algorithm was enabled with a time window of 2 min to transfer identifications between adjacent gel slices, only for samples analysed using the same nanospray conditions. Database searching was performed using the Andromeda search engine integrated into the MaxQuant environment against the mouse Uniprot proteome database, concatenated with known contaminants and reversed sequences of all entries. Protein, peptide and site FDR was controlled at a maximum of 1% respectively. All contaminants and reverse sequenced peptides were removed, and the data were log2 transformed. This lead to the identification of 5908 protein groups, of which 2620 correspond to unique proteins and 3288 contained in multiple proteins. We subsequently applied Re-Fraction (*Yang et al., 2012*) on protein groups that contained multiple proteins and resolved a further 960 unique protein identifications. For groups that could not be identified as a single protein, proteins with the lowest posterior error probability were selected.

For human adipose tissue experiments, raw mass spectrometry data were processed using the MaxQuant software (*Cox and Mann, 2008*; *Cox et al., 2009*) version 1.5.7.0 using the default settings with minor changes: Oxidised Methionine (M) and Acetylation (Protein N-term) were selected as variable modifications, and carbamidomethyl (C) as fixed modification. A maximum of two missed cleavages was permitted, 10 peaks per 100 Da, MS/MS tolerance of 20 ppm, and a minimum peptide length of 6. The 'matching between runs' algorithm was enabled with a time window of 2 min to transfer identifications between adjacent samples. Database searching was performed using the Andromeda search engine (*Cox et al., 2011*) integrated into the MaxQuant environment against the human Uniprot proteome database, concatenated with known contaminants and reversed sequences of all entries. Protein, peptide and site FDR was controlled at a maximum of 1% respectively. All contaminants and reverse sequenced peptides were removed, and the data were log2 transformed. To normalise the protein quantifications across different samples, proteins were first ranked within each sample based on their MS intensities and then ranked across all samples to identify the most stable proteins for data normalisation. Using this approach, 40S ribosomal protein (RPSA) was selected for normalisation by calculating a log fold change of protein intensities with respect to RPSA within each sample, allowing relative protein abundance (refer to as normalised ratio) to be comparable across all samples for all proteins.

## Microarray analysis

RNA was prepared using the RNeasy protocol (Qiagen, Valencia, CA). Quantity and quality of total RNA samples were determined using an ND-1000 spectrophotometer (Thermo Fisher Scientific) and Bioanalyzer 2100 (Agilent Technologies, Palo Alto, CA), respectively. RNA with RNA integrity numbers > 8, 260/280 values > 1.8 and 260/230 values > 1.8 was considered acceptable. Labelled cRNA was hybridised to Mouse Genome 430 2.0 arrays. Initial data analysis files were generated using manufacturer's GeneChip Command Console Software. Relative mRNA contents were studied for 45,000 transcripts for cells and tissue.

## Microarray data processing

All statistical data analysis for the gene expression data were performed in the R-programming environment. AT transcripts in the mouse4302 Affymetrix array was pre-processed using R package affyPLM (*Bm, 2004*; *Bolstad BM et al., 2005*). Background correction, quantile normalisation and summarisation was performed using RMA function (*Irizarry et al., 2003*). Quality control assessments of the arrays were performed and problematic arrays were identified and down-weighted for ensuing statistical inference tests using function arrayWeights (*Ritchie et al., 2006*) within LIMMA (*Ritchie et al., 2015*) during linear model fitting of the microarray data. Probes were collapsed to gene level using the maximal average expression of probes across all samples.

## Data availability

The mass spectrometry proteomics data have been deposited to the ProteomeXchange Consortium via the PRIDE (*Vizcaíno et al., 2016*) partner repository with the dataset identifiers PXD005128 and PXD006891. The microarray discussed in this manuscript have been deposited in NCBI's Gene

Expression Omnibus (*Edgar et al., 2002*) and are accessible through GEO Series accession numbers GSE87853 and GSE87854.

## Differential transcript and protein analysis

Differential expression analysis of proteins and transcripts within each model of insulin resistance were performed using moderated t-test from LIMMA package (*Goeman and Bühlmann, 2007*; *Michaud et al., 2008*; *Wu and Smyth, 2012*). For the cell models and adipose tissue, proteins and transcripts with an absolute fold change (FC) >1.5 and controlled for 5% FDR were considered differentially expressed. Empirical Bayes was used for global variance shrinkage and Benjamini and Hochberg method (*Benjamini and Hochberg, 1995*) was used to correct multiple hypothesis testing. The moderated t-statistics from differential expression analysis in each model was transformed to z-scores and used as the test statistics for the direction analysis (*Yang et al., 2014*) to identify genes/proteins altered (up/down) across multiple experimental conditions. For the cell models, we performed a 3-dimensional direction analysis (across CI, Dex and TNF) whereas in AT, we performed a 2-dimensional direction analysis (across 5 and 14 d of HFHSD). Significance for transcripts was defined as p-value<0.001 and for proteins at 0.01.

## Pathway enrichment analysis

Pathway enrichment for each condition in cell models and adipose tissue were analysed using by testing up- and down-regulation of 192 pathways from the KEGG database (*Kanehisa and Goto, 2000*) as curated by the Molecular Signature Database C2: curated gene sets (*Liberzon et al., 2011*; *Subramanian et al., 2005*) using the mean-rank gene-set enrichment test implemented in LIMMA R package (*Goeman and Bühlmann, 2007*; *Michaud et al., 2008*; *Wu and Smyth, 2012*) in our transcript and protein data. The fold change (log2 scale) of genes and proteins from the individual DE analysis in each model was used as the test-statistics for the GSEA analysis. Significance of pathways was defined at p-value<0.05. CoQ pathway members were edited from the original KEGG annotation to include recent annotations and consisted of Hpd, Tat, Coq2/3/4/5/6/7/9, Adck3, and Adck4.

## Pathway integrative analysis and phenotype association

For human proteome data, we associated the normalised ratio of each protein with phenotypical information across all samples using pairwise Pearson's correlation. To perform a similar analysis on pathway level, for each protein, we calculated a relative protein ratio in each sample by subtracting the average of normalised ratios across all samples. Subsequently, for each KEGG pathway, we summarised its overall trend within each sample by summing the relative ratios of all proteins belong to that pathway. The summary statistics for each pathway in each sample was then used to associate with phenotypical information using pairwise Pearson's correlation. These results were subsequently visualised as heatmaps.

To identify pathways enriched across multiple tissue and/or cell models, we used Fisher's combined statistics to integrate p-values from individual pathway enrichment analysis conducted on each condition in cell models and adipose tissue. Fisher's combined statistics follows a chi-square distribution and the integrated p-values from the chi-square distribution are converted into z-scores (referred to as combined z-scores) for cell model and adipose tissue, respectively, to visualise each pathway and their overall enrichment across multiple conditions in cell models and tissue.

## Determination of cholesterol and CoQ

$CoQ_9$ and $CoQ_{10}$ content in 3T3-L1 adipocyte lysates, tissue homogenates, and membrane fractions from cells, and tissue were determined as described previously (*Gay and Stocker, 2004*). Investigators were blinded to experimental groups. Briefly, samples were thawed, mixed gently and 50–450 µL placed in a 15 mL screw top tube to which 2 mL methanol and 10 mL of water-washed hexane were added. The mixture was then mixed vigorously for 1 min, centrifuged (1430 x g, 5 min, 4°C) and 9 mL of the top hexane layer collected then dried using a rotary evaporator. The resulting dried lipids were re-dissolved in 180 µL ice-cold mobile phase (ethanol:methanol:isopropanol: ammonium acetate pH 4.4, 65:30:3:2, vol/vol/vol/vol) and transferred into HPLC vials. Cholesterol, $CoQ_9$, $CoQ_9H_2$, $CoQ_{10}$ and $CoQ_{10}H_2$ were determined by HPLC using a Supelcosil LC-C18 column (5 µm,

250 × 4.6 mm) eluted at 1 mL/min and connected to UV and electrochemical (ESA CoulArray 5600A) detectors. NEC was detected at 214 nm, while $CoQ_9$, $CoQ_9H_2$, $CoQ_{10}$ and $CoQ_{10}H_2$ were detected at −700, +700 and +500 mV and quantified against authentic commercial standards obtained from Sigma Aldrich (USA). $CoQ_9H_2$ and $CoQ_{10}H_2$ standards were generated by sodium borohydride-mediated reduction of commercial $CoQ_9$ and $CoQ_{10}$, respectively.

For accurate determination of $CoQ_9$ and $CoQ_9H_2$ for calculation of CoQ redox status analyses were performed as above with several modifications to sample preparation to limit oxidation. Cells were washed in PBS containing 100 µM diethylenetriaminepentaacetic acid (DTPA) and homogenised in HES buffer pre-gassed with argon. Homogenisation was carried out under an argon atmosphere. Homogenates were processed one-at-a-time and samples dried in the rotary evaporator under argon. Samples were analysed immediately.

Liquid chromatography/tandem mass spectrometry was used for the determination of $CoQ_9$, $CoQ_9H_2$, $CoQ_{10}$ and $CoQ_{10}H_2$ in human and mouse adipose tissue after in vivo supplementation with CoQ10. Briefly, to 50–450 µL of adipose tissue homogenates, internal standard (200 pmol $CoQ_8$; Avanti Polar Lipids) were added and homogenates extracted with methanol/hexane as described above. Following evaporation of the hexane phase, the resulting dried lipids were re-dissolved in 150 µL ice-cold ethanol (HPLC grade) and 5 µL injected onto an Agilent 1290 UHPLC system connected to an Agilent 6490 triple-quadrupole mass spectrometer. Analytes were separated on a 2.6 µm Kinetex XB-C18 100 A column (50 × 2.10 mm; Phenomenex, USA) by gradient elution using mobile phase A (2.5 mM ammonium formate in 95:5 methanol:isopropanol) and mobile phase B (2.5 mM ammonium formate in 100% isopropanol) at 0.8 mL/min. The gradient consisted of 0% mobile phase B from 0 to 1.5 min, 0–10% B from 1.5 to 2 min, 10% B from 2 to 3 min and back to 0% B from 3 to 5 min. Flow was then directed into the triple quadrupole mass spectrometer with parameters set as follows: gas temperature = 250℃; gas flow = 20 L/min; nebuliser pressure = 35 psi; sheath gas heater = 325℃; sheath gas flow = 12 L/min; capillary voltage = 3,500 V. Detection of $CoQ_8$, $CoQ_9$, $CoQ_9H_2$, $CoQ_{10}$ and $CoQ_{10}H_2$ was by multiple reaction monitoring (MRM) in positive ion mode using the above general mass spectrometry parameters with fragmentor voltage at 380 V and cell accelerator voltage at 5 V. In each case, the fragment ions generated by collision-induced dissociation of the $[M + H]^+$ or the $[M+NH_4]^+$ ions were used for quantification. MRM settings for the target analytes were (parent ion → fragment ion); $CoQ_8$ (m/z 727.1 → 197.1) with collision energy (CE) = 33 V; $CoQ_9$ (m/z 795.5 → 197.1) with CE = 33 V; $CoQ_9H_2$-$NH_4$ (m/z 816.6 → 197.1) with CE = 25 V; $CoQ_{10}$ (m/z 863.6 → 197.1) with CE = 37 V; and $CoQ_{10}H_2$ (m/z 882.7 → 197.1) with CE = 33 V. $CoQ_8$, $CoQ_9$ and $CoQ_{10}$ were quantified against authentic commercial standards obtained from Sigma Aldrich (USA), while $CoQ_9H_2$ and $CoQ_{10}H_2$ standards were generated from $CoQ_9$ and $CoQ_{10}$, respectively by sodium borohydride-mediated.

For determination of $CoQ_9$ synthesis rates, cells were incubated with 50 µM $^{13}C_6$-4-hydroxybenzoic acid (Cambridge isotope, USA) in DMEM containing GlutaMAX and dialysed FCS for 12 hr following treatment to induce insulin resistance. Cells were washed with cold PBS and homogenised in HES-I buffer (10 mM HEPES, pH 7.4, 1 mM EDTA, 250 mM sucrose, protease inhibitors) by 12 strokes of a Dounce homogeniser. To 350 µL of cell homogenates, internal standard (200 pmol $CoQ_8$; Avanti Polar Lipids) was added and homogenates extracted with methanol/hexane as described above. Following evaporation of the hexane phase, the resulting dried lipids were re-dissolved in 150 µL ice-cold ethanol (HPLC grade) and 5 µL injected onto an Agilent 1290 UHPLC system connected to an Agilent 6490 triple-quadrupole mass spectrometer with column, mobile phases, gradient elution, flow rate and mass spectrometry parameters as above. MRM settings for $^{13}C_6$-$CoQ_9$ (parent ion → fragment ion) were m/z 801.5 → 203 with collision energy (CE) = 33 V. $^{13}C_6$-$CoQ_9$ was quantified against authentic $CoQ_9$ commercial standard.

## Subcellular fractionation

To measure cholesterol and $CoQ_9$ content is different cellular compartments (*Figure 3* and *Figure 3—figure supplement 1*), we performed isolated plasma membrane, mitochondrial and microsomes by differential centrifugation. 3T3-L1 adipocytes were washed with ice-cold PBS, harvested in ice-cold HES-I, and subsequent steps were carried out at 4℃. Cells were homogenised by 12 strokes of a Dounce homogeniser to yield a whole cell homogenate prior to centrifugation at 700 × g for 10 min to pellet nuclei and unbroken cells. The resulting supernate was centrifuged at 13,550 × g for 12 min to pellet the plasma membrane (PM) and mitochondria, with the resulting supernate

consisting of cytosol and microsomes. This supernate was then centrifuged at 235,200 × g for 75 min to obtain a cytosol fraction (supernate) and a microsomal fraction (pellet). The PM/mitochondria pellet was re-suspended/washed in HES-I and re-centrifuged at 13,550 × g for 12 min. The PM/mitochondria pellet was re-suspended again in HES-I and layered over a high sucrose HES buffer (1.12 M sucrose, 0.05 mM EDTA, 10 mM HEPES, pH 7.4) and centrifuged at 111,160 × g for 60 min in a swing-out rotor. The pellet was the mitochondria fraction. The PM fraction was collected above the sucrose layer, and centrifuged again at 13,550 × g for 12 min to achieve a PM pellet. All fractions were re-suspended in HES-I. Protein concentration for each fraction was performed using the BCA assay.

## Mitochondrial isolation

For specific assessment of mitochondrial CoQ content in cell culture (*Figure 4*), and for assessment of mitochondrial $CoQ_9$ or $CoQ_{10}$ in adipose tissue, skeletal muscle and liver from mice and adipose tissue from humans we used a protocol to directly enrich mitochondria (*Figure 3*, *Figure 3—figure supplement 1*). Mitochondrial isolation of cultured adipocytes and tissues from mice and humans was performed as previously described (*Frezza et al., 2007*). Briefly, cells/tissue was homogenised in mitochondrial isolation buffer (10 mM Tris-MOPS, pH 7.4, 1 mM EGTA, 200 mM sucrose containing protease inhibitors) by 12 strokes of a Dounce homogeniser at 4°C and samples kept at 4°C subsequently. Homogenates were centrifuged at 700 x g for 10 min and then the supernate centrifuged at 7000 x g for 10 min to obtain a pellet containing the mitochondria. The pellet was re-suspended in mitochondrial isolation buffer and re-centrifuged at 7000 x g for 10 min. The mitochondrial pellet was finally re-suspended in mitochondrial isolation buffer and protein concentration determined using the BCA assay.

## Citrate synthase assay

Cultured adipocytes or adipose tissue was homogenised in mitochondrial extraction buffer (50 mM potassium phosphate, 0.5 mM EGTA, 0.1% Triton, pH 7.4) by 12 strokes of a Dounce homogeniser. Homogenates were freeze-thawed three times and insoluble debris pelleted by centrifugation at 5000 x g for 10 min. Citrate synthase activity was assayed by adding diluted samples to citrate synthase assay buffer (100 mM Tris:HCl, pH 8, 100 mM $MgCl_2$ 50 mM EDTA) containing 0.5 mM dithionitrobenzoic acid and 80 µM acetyl-CoA and the reaction monitored spectrophotometrically at 37°C at 412 nm for background activity. Oxaloacetate was added to 1 mM to start the reaction. Background activity prior to the addition of oxaloacetate was subtracted to obtain rates specific to citrate synthase. Citrate synthase activity was calculated as mU/mg using the extinction coefficient dithionitrobenzoic acid for (ε = 13.6 µmol/mL/cm) and taking sample dilution into account.

## Cell lysis

Cells were washed twice with PBS and lysed in 1% SDS in PBS-containing protease inhibitors. Cell lysates were sonicated (Bandelin SONOPLUS) for 12 s and centrifuged at 13,000 × g for 15 min at room temperature. The protein concentration of the resulting supernate was then determined by the BCA assay.

## SDS-PAGE and western blotting

Proteins (5–10 µg) from cell/tissue lysates or homogenates were resolved by SDS-PAGE and transferred to polyvinylidene difluoride membranes. Membranes were blocked in 5% BSA or skim milk powder in Tris-buffered saline containing 0.1% (v/v) Tween 20 for 1 hr, followed by an overnight incubation at 4°C with specific primary antibody solutions. Membranes were incubated with an appropriate secondary antibody for 1 hr before signals were detected using ECL (Thermo Fisher Scientific or Millipore) on the Chemidoc MP (Bio-Rad) or on the Odyssey Fluorescence Detection System (LiCOR). Antibodies detecting multiple mitochondrial complex subunits (Cat. No. 45–8099) and PRDX1 were from Thermo Fisher Scientific (Cat. No. PA3-750), anti-PRDX2 (Cat. No. ab109367, clone: EPR5154) and anti-catalase (Cat. No. ab52477) antibodies were from Abcam, anti-PRDX3 antibody from Ab Frontier (Cat. No. LF-PA0030), anti-14-3-3 antibody was from Santa Cruz (Cat. No. sc-629, clone K19), anti-pT642 TBC1D4 (Cat. No. 4288), anti- TBC1D4 (Cat. No. 2670), anti-pT308 Akt (Cat. No. 9275), anti-pS473 Akt (Cat. No. 4051) and anti-Akt (Cat. No. 4685) antibodies were from

Cell Signaling Technologies and anti-α-tubulin antibody was from Sigma Aldrich (Cat. No. T9026). Antibody to GLUT4 was generated in-house. Densitometry analysis was performed using Image Lab 5.2.1 (Bio-Rad) or Image Studio (LiCOR).

## In vivo administration of CoQ

For studies in chow and HFHSD-fed mice, male C57BL/6J (n = 9 per treatment) fed *ad libitum* with normal chow or HFHSD for a period of 14 d. Mice were treated with 10 mg/kg the liposomal $CoQ_{10}$ (LiQsorb, Tichson Corp.) diluted in saline by intraperitoneal injection every 48 hr for the duration of the feeding regimen. Control mice received saline. Mice were randomly assigned to a treatment group so groups were equally represented within each cage.

For dose response studies, male C57BL/6J (n = 5 per treatment) fed *ad libitum* for a period of 14 d with HFHSD were treated with the specified dose of liposomal $CoQ_{10}$ (LiQsorb, Tichson Corp.) (0, 1, 5 or 10 mg/kg) diluted in saline by intraperitoneal injection every 48 hr for the duration of the feeding regimen. Saline was given to the control group. Mice were randomly assigned to a treatment group so that at least one mouse per cage was in each group. For each mouse, epididymal fat pads were assessed for mitochondrial content of $CoQ_9$ and $CoQ_{10}$, PRDX3 dimer/monomer status and insulin sensitivity by the ex vivo 2DOG uptake assay. For ex vivo 2DOG uptake assay, saline-injected chow mice were included as a control.

## Inhibition of mevalonate pathway or CoQ synthesis

Differentiated 3T3-L1 adipocytes were treated with 10 µM simvastatin or 20 µM atorvatstin for indicated times or 2.5 mM 4-nitrobenzoic acid (*Alam et al., 1975*; *Forsman et al., 2010*) or 4-chlorobenzoic acid (*Bour et al., 2011*) for 48 hr. These doses were selected to lower mitochondrial CoQ content to that observed in insulin-resistant 3T3-L1 adipocytes.

## siRNA-mediated knockdown of *Coq7* or *Coq9*

Seven days post-differentiation, adipocytes were trypsinised (5 × trypsin, EDTA) (Thermo Fisher Scientific) at 37°C, washed twice with PBS, and re-suspended in electroporation solution (20 mM HEPES, 135 mM KCl, 2 mM $MgCl_2$, 0.5% Ficoll 400, 1% DMSO, 2 mM ATP, and 5 mM glutathione, pH 7.6) with 200 nM scrambled (sense 5′-CAGTCGCGTTTGCGACTGGTT-3′) or pooled anti-*Coq7* siRNA (sense 5′-GGGAUCACGCUGGUGAAUAUTT-3′, 5′-GGAUGACCUUAGACAAUAUTT-3′, 5′-GCCUUGUUGAAGAGGAUUAUTT-3′) or pooled anti-*Coq9* siRNA (sense 5′-GCAGCAUUCUGAGA-CACAGTT-3′, 5′-GCUGGUGAUGAUGCAGGAUTT-3′, 5′- GCAAUGAACAUGGGCCACATT-3′) or (Shanghai Genepharma). Cells were electroporated at 200 mV for 20 ms using an ECM 830 square wave electroporation system (BTX Molecular Delivery Systems) and seeded onto Matrigel (Corning)-coated plates. Cells were assayed 96 hr following electroporation.

## Sample preparation and real-time quantitative-PCR assays

Total RNA was extracted from cells using TRIzol reagent (Thermo Fisher Scientific). cDNA synthesis was carried out using PrimeScript first strand cDNA synthesis kit (Clontech and Takara Bio Company). Polymerase chain reactions were performed on the LightCycler 480 II (Roche Applied Science) using FastStart Universal SYBR Green Master (Roche Applied Science). Cyclophilin b was used as an internal control. The primer sets used were as follows: mCoq7_F; tttggaccatagctgcattg and mCoq7_R; tgaggcctcttccatactctg, mCoq9_F; tcagcagcattctgagacaca and mCoq9_R; gtgctgtagctgctcctcact, and mCypB-F; ttcttcataaccacagtcaagacc; mCypB-R, accttccgtaccacatccat.

## Mitochondrial bioenergetics in intact cells

The bioenergetics of intact cells was assessed as described previously (*Krycer et al., 2017*). Briefly, On Day 7–8 of differentiation, 3T3-L1 adipocytes were trypsinised with 5x Trypsin/EDTA (Thermo Fisher Scientific) in phosphate-buffered saline (PBS) and seeded onto XFp cell culture plates coated with Matrigel (Corning, distributed by Sigma Aldrich (Castle Hill, NSW, Australia)). Following the indicated treatment, cells were washed three times with warm PBS, once with bicarbonate-free DMEM buffered with 30 mM Na-HEPES, pH 7.4 (DMEM/HEPES), and then incubated in DMEM/HEPES supplemented with 0.2% (w/v) BSA, 25 mM glucose, 1 mM GlutaMAX and 1 mM glutamine, for 1.5 hr in a non-CO2 incubator at 37°C. Cells were then washed once with PBS, once with DMEM/

HEPES, and assayed in the XFp Analyzer in DMEM/HEPES with 25 mM galactose, 1 mM GlutaMAX and 1 mM glutamine and 1 mM NaHCO$_3$. During the assay, respiration was assayed with mix/wait/read cycles of 3/0/2 min for 3T3-L1 adipocytes. Following assessment of basal respiration, the following compounds were injected sequentially: oligoymcin (10 µg/ml), BAM15 (10 µM) (*Kenwood et al., 2014*) rotenone/antimycin A (5 µM / 10 µM). All of these reagents were obtained from Sigma Aldrich, except BAM15 (*Kenwood et al., 2014*). After the assay, the medium was aspirated and DNA content was measured by Hoechst staining as described previously (*Krycer et al., 2017*).

## Mitochondrial bioenergetics in permeabilised cells

3T3-L1 adipocytes were seeded into Matrigel-coated XFp cell culture plates as described above. Following the indicated treatment, cells were washed with warm PBS and incubated in mitochondrial respiration buffer (20 mM K-HEPES pH 7.1, 0.5 mM EGTA, 3 mM MgCl$_2$, 10 mM KH$_2$PO$_4$, 110 mM D-sucrose, 0.1% (w/v) fatty-acid free BSA) for 30 min at 37°C. Cells were permeabilised with 10 µg/mL digitonin prior to equilibration in the Seahorse XFp Analyzer (Seahorse Biosciences). The following injections were used: Port A: ADP-regeneration system (5 mM deoxyglucose, 1 U/mL hexokinase (Roche), 1 mM ADP), 2 mM malate and 10 mM glutamate; Port B: 10 µM rotenone; Port C: 10 mM succinate or 100 µM octanoylcarnitine; Port D: 10 µM antimycin A, 5 mM ascorbic acid and 0.5 mM tetramethyl-phenylenediamine. Each cycle included 3 min mixing time, 0 min waiting time and 2 min measuring time, repeated until a steady $JO_2$ reading was obtained after each injection. Non-mitochondrial respiration was taken as $JO_2$ following rotenone addition and this was subtracted from all substrate-driven rates (ports A, C and D). State four respiration rates (proton leak) was determined by omission of the ADP-regeneration system. All of these reagents were obtained from Sigma Aldrich, unless otherwise stated. The resulting rate was normalised to DNA content, measured using Hoechst 33248 with a previously-described Hoechst-staining protocol (*Krycer et al., 2017*).

## Assessment of peroxiredoxin dimerisation

Sample preparation was carried out essentially as previously described (*Bayer et al., 2013*). For in vitro studies, 3T3-L1 adipocytes were pre-treated as indicated and washed three times with ice-cold PBS that had been pre-treated with 10 µg/mL catalase for 1 hr. Cells were incubated with 100 mM *N*-ethylmaleimide in PBS for 10 min on ice to modify free cysteine residues. Cells were scraped in PBS-containing 1% SDS, protease inhibitors and 100 mM *N*-ethylmaleimide. For tissue preparation, epididymal adipose tissue was immediately lysed in 2 x RIPA buffer containing protease inhibitors and 100 mM *N*-ethylmaleimide upon collection buffer by sonication (20 s). Samples were centrifuged at 13,000 × g for 10 min at room temperature. Fat was removed, supernatant collected and protein concentration determined by BCA assay (Thermo Fisher Scientific). Proteins were then separated by non-reducing SDS-PAGE. Intensities of PRDX dimer and monomer bands were determined by densitometry and ratios (dimer/monomer) expressed relative to control samples within each experiment.

## Assessment of conversion of Mito-hydroethidine (MitoSOX) to Mito-2-hydroxyethidium

Conversion of MitoSOX to Mito-2-hydroxyethidium was detected by LC-MS/MS, as previously described (*Zielonka et al., 2008*) with modifications. 3T3-L1 adipocytes were treated as indicated and incubated with 5 µM MitoSOX (Thermo Scientific) for 60 min in DMEM/GlutMAX without FCS. Cells were scraped in ethanol and centrifuged at 17,000x g for 15 min, 4°C. 5 µL of supernate were then injected onto an Agilent 1290 UHPLC system connected to an Agilent 6490 triple-quadrupole mass spectrometer. Analytes were separated on a 2.6 µm Kinetex XB-C18 100 A column (50 × 2.10 mm; Phenomenex) by gradient elution using mobile phase A (0.1% formic acid in water) and mobile phase B (0.1% formic acid in acetonitrile) at 0.4 mL/min. The gradient consisted of 25% mobile phase B from 0 to 2 min, 25–40% B from 2 to 7 min, 40–90% B from 7 to 8 min, 90% B from 8 to 11 min and back to 25% B from 11 to 12 min. Flow was then directed into the triple quadrupole mass spectrometer with parameters set as follows: gas temperature = 290°C; gas flow = 14 L/min; nebulizer pressure = 35 psi; sheath gas heater = 350°C; sheath gas flow = 11 L/min; capillary voltage = 3,500 V. Detection of MitoSOX, Mito-ethidium (Mito-E+) and Mito-hydroxyethidium (Mito-2OHE+) was by multiple reaction monitoring (MRM) in positive ion mode using the above general mass spectrometry parameters with fragmentor voltage at 380 V and cell accelerator voltage at 5 V. In each case,

the fragment ions generated by collision-induced dissociation of the [M + H]+or the [M + 2 hr]+ ions were used for quantification. MRM settings for the target analytes were (parent ion → fragment ion); MitoSOX (m/z 632.2 → 289) with collision energy (CE) = 25 V; Mito-E+ (m/z 315.7 → 289) with CE = 25 V and Mito-2OHE+ (m/z 323.7 → 289) with CE = 25 V. MitoSOX was quantified against authentic commercial standard obtained from Invitrogen while Mito-E+ and Mito-2OHE+ were quantified against standards synthesized from MitoSOX as previously described (*Zielonka et al., 2008*).

## Treatment of adipocytes with mitochondrial poisons

3T3-L1 adipocytes were treated as indicated prior to treatment with 25 µM BAM15, 10 µM FCCP (Sigma Aldrich), 10 µM rotenone (Sigma Aldrich), 50 nM antimycin A (Sigma Aldrich), 10 µg/mL oligomycin (Sigma Aldrich), 10 or 100 µM TTFA (Sigma Aldrich) or 10 mM malonate (Sigma Aldrich) for 2 hr in DMEM/10% FCS/GlutaMAX before determination of PRDX dimerisation status. For assessment of the effect of TTFA or malonate on HA-GLUT4 translocation in 3T3-L1 adipocytes, cells were incubated with 100 µM TTFA or 10 mM malonate for 2 hr in DMEM/10% FCS/GlutaMAX and for 2 hr during serum starvation in DMEM/0.2% BSA/GlutaMAX prior to assessment of insulin action.

## Overexpression of mitochondrial-targeted catalase (mCat)

Mitochondrial-targeted human catalase (human catalase with the 24 residue MnSOD mitochondrial localisation signal at its N-terminus) (*Arita et al., 2006*) was inserted into the pBABE-puro vector. Mitochondrial-targeted catalase overexpressing 3T3-L1 fibroblasts were generated by retroviral transduction as previously described (*Govers et al., 2004*). 3T3-L1 fibroblasts were transduced and selected using puromycin. Fibroblasts were differentiated into adipocytes as described.

## Statistical analyses

Data are expressed as means ± S.E.M. For each dataset, the sub-groups were initially compared with the relevant ANOVA depending on the levels of treatments. If row or column factors were significant, specific sub-groups were then compared using Student's *t*-test (adjusting for multiple comparisons using the Sidak method). Significant effects were defined as $p < 0.05$ by these t-tests, as reported in the figures. Human data were assessed by Mann Whitney tests and significant effects were defined as $p < 0.05$.

## Acknowledgements

We thank G Dallner for advice on pharmacological inhibition of CoQ biosynthesis. JRK and BLP are National Health and Medical Research Council (NHMRC) of Australia Early Career Fellows. DJF held a Sir Henry Wellcome fellowship for the majority of this study. RS and DEJ hold NHMRC Senior Principal Research Fellowships. This work was supported by NHMRC of Australia Project Grants 1061122 and 1086850 awarded to DEJ, DP150102408 from the Australian Research Council to RS and NHMRC Program Grant 1052616 to RS. The contents of the published material are solely the responsibility of the authors and do not reflect the views of the NHMRC of Australia.

## Additional information

### Competing interests

Ganesh Kolumam, Jean YH Yang: Employed by Genentech Inc. at the time the study was conducted. The other authors declare that no competing interests exist.

### Funding

| Funder | Grant reference number | Author |
| --- | --- | --- |
| Wellcome | 092255/Z/10/Z | Daniel J Fazakerley |
| National Health and Medical Research Council | 1052616 | Roland Stocker |
| Australian Research Council | DP150102408 | Roland Stocker |

| National Health and Medical Research Council | 1086850 | David E James |
| National Health and Medical Research Council | 1061122 | David E James |

The funders had no role in study design, data collection and interpretation, or the decision to submit the work for publication.

## Author contributions

Daniel J Fazakerley, Conceptualization, Data curation, Formal analysis, Supervision, Funding acquisition, Validation, Investigation, Visualization, Writing—original draft, Project administration, Writing—review and editing; Rima Chaudhuri, Pengyi Yang, Data curation, Software, Formal analysis, Visualization, Writing—review and editing; Ghassan J Maghzal, Supervision, Investigation, Project administration, Writing—review and editing; Kristen C Thomas, Validation, Investigation, Project administration, Writing—review and editing; James R Krycer, Benjamin L Parker, Kelsey H Fisher-Wellman, Nolan J Hoffman, Ciana Diskin, James G Burchfield, Daniel L Chen, Investigation, Writing—review and editing; Sean J Humphrey, Conceptualization, Investigation, Visualization, Writing—review and editing; Christopher C Meoli, Investigation Writing—review and editing; Mark J Cowley, Warren Kaplan, Software, Formal analysis, Writing—review and editing; Zora Modrusan, Ganesh Kolumam, Resources, Investigation, Writing—review and editing; Jean YH Yang, Supervision, Writing—review and editing; Dorit Samocha-Bonet, Formal analysis, Supervision, Investigation, Writing—review and editing; Jerry R Greenfield, Resources, Supervision, Writing—review and editing; Kyle L Hoehn, Conceptulization, Investigation, Writing—review and editing; Roland Stocker, David E James, Conceptualization, Resources, Supervision, Funding acquisition, Writing—original draft, Project administration, Writing—review and editing

## Author ORCIDs

Daniel J Fazakerley (iD) https://orcid.org/0000-0001-8241-2903
David E James (iD) https://orcid.org/0000-0001-5946-5257

## Ethics

Human subjects: The study protocol was approved by St Vincent's Hospital Human Research Ethics Committee (Sydney, Australia) (HREC/10/SVH/133) and written consent obtained prior to the study. Animal experimentation: All experiments were carried out with the approval of the Garvan Institute/ St. Vincent's Hospital Animal Experimentation Ethics Committee (09/46) or the approval of the University of Sydney Animal Ethics Committee (2014/694), following guidelines issued by the National Health and Medical Research Council of Australia.

## Decision letter and Author response

Decision letter https://doi.org/10.7554/eLife.32111.031
Author response https://doi.org/10.7554/eLife.32111.032

# Additional files

## Supplementary files

• Supplementary file 1. Proteomic analysis of insulin-resistant mouse adipose tissue and 3T3-L1 adipocytes. (A-D) Differential expression and directional analyses of proteomic data from adipose tissue (A, B) and 3T3-L1 adipocytes (C, D). Values in directional analysis sheets are p values. E, Pathway analysis of tissue and cell proteomic data.
DOI: https://doi.org/10.7554/eLife.32111.018

• Supplementary file 2. Transcriptomic analysis of insulin-resistant mouse adipose tissue and 3T3-L1 adipocytes. (A-D) Differential expression and directional analyses of transcript data from adipose tissue (A, B) and 3T3-L1 adipocytes (C, D). Values in directional analysis sheets are p values. E, Pathway analysis of tissue and cell transcript data.
DOI: https://doi.org/10.7554/eLife.32111.019

• Supplementary file 3. Correlation of the human adipose tissue proteome with clinical traits. Correlation of protein (A) and pathway (B) expression with specified clinical measures. Significant r value = <−0.423 or >0.423. Tables contain combined z-score for proteins and pathway from in vivo and in vitro analyses.
DOI: https://doi.org/10.7554/eLife.32111.020

• Transparent reporting form
DOI: https://doi.org/10.7554/eLife.32111.021

### Major datasets

The following datasets were generated:

| Author(s) | Year | Dataset title | Dataset URL | Database, license, and accessibility information |
|---|---|---|---|---|
| Rima Chaudhuri, Zora Modrusan, David E James | 2016 | Insulin resistance in 3T3-L1 adipose cells | https://www.ncbi.nlm.nih.gov/geo/query/acc.cgi?acc= GSE87853 | Publicly available at the NCBI Gene Expression Omnibus (accession no. GSE87853) |
| Rima Chaudhuri, Zora Modrusan, David E James | 2016 | Insulin resistance in high fat diet mouse (adipose) | https://www.ncbi.nlm.nih.gov/geo/query/acc.cgi?acc=GSE87854 | Publicly available at the NCBI Gene Expression Omnibus (accession no. GSE87854) |
| Daniel J Fazakerley, David E James | 2017 | Human adipose insulin resistance | https://www.ebi.ac.uk/pride/archive/projects/PXD006891 | Publicly available at EBI PRIDE (accession no. PXD006891) |
| Daniel J Fazakerley, David E James | 2016 | Proteomic analysis of adipocyte insulin resistance | https://www.ebi.ac.uk/pride/archive/projects/PXD005128 | Publicly available at EBI PRIDE (accession no. PXD005128) |

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
