## [Decision Letter]

Thank you for submitting your article "Mitochondrial CoQ deficiency is a common driver of oxidative stress and insulin resistance" for consideration by *eLife*. Your article has been reviewed by three peer reviewers, one of whom is a member of our Board of Reviewing Editors and the evaluation has been overseen by Mark McCarthy as the Senior Editor. The reviewers have opted to remain anonymous.

The reviewers have discussed the reviews with one another and the Reviewing Editor has drafted this decision to help you prepare a revised submission.

Summary:

James et al. use a proteomics approach to screen for novel pathways involved in insulin resistance. By cross-referencing results obtained from 3T3-L1 cell line insulin resistance models with a sucrose-fed mouse model they identify changes to several pathways. This is an ambitious and comprehensive effort. The authors identify the mevalonate-Coenzyme Q pathway as a key pathway and propose a causative mechanism to explain the correlation. Although most of the studies presented are supportive of the hypothesis, the evidence supporting a causal relationship between loss of CoQ biosynthesis and the development of insulin resistance is not definitive.

The reviewers are in agreement that the work was potentially of interest to the general audience of *eLife*. However, they believe that additional studies are required to support the idea that CoQ supplementation or manipulation of components of this pathway can improve insulin sensitivity, especially since this hypothesis was previously tested, and while some studies report improvement of hyperglycemia, others have reported no effect.

Essential revisions:

1) The authors focus primarily on glucose uptake to evaluate insulin responses. While this makes sense for skeletal muscle, for adipose tissue (the primary tissue studied in this manuscript) the authors should test the effects on insulin-mediated suppression of lipolysis. This is especially important since NEFA liberation from fat is a major contributor to hepatic glucose production. This could be assessed by measuring glycerol/FFA release to the media following the same manipulations described in the manuscript that reduce/enhance insulin-stimulated glucose uptake. Measurements of serum FFA can also be done following CoQ treatment and glucose injection.

2) Along the same lines, since the authors claim that CoQ supplement improves insulin sensitivity, it would be important to perform ITTs to support this.

3) The improvement in insulin sensitivity upon catalase overexpression and complex II inhibition seems modest (Figure 6, Figure 7). To strengthen the causality between CoQ levels and insulin resistance it would be helpful to show whether TTFA/malonate treatment also improves insulin sensitivity in adipose explants derived from HFHSD-fed mice.

---

## [Author Response]

1) The authors focus primarily on glucose uptake to evaluate insulin responses. While this makes sense for skeletal muscle, for adipose tissue (the primary tissue studied in this manuscript) the authors should test the effects on insulin-mediated suppression of lipolysis. This is especially important since NEFA liberation from fat is a major contributor to hepatic glucose production. This could be assessed by measuring glycerol/FFA release to the media following the same manipulations described in the manuscript that reduce/enhance insulin-stimulated glucose uptake. Measurements of serum FFA can also be done following CoQ treatment and glucose injection.

These data are included in the revised manuscript.

We assessed insulin-mediated suppression of lipolysis in all in vitro models of insulin resistance including NB-treated cells and cells in which *Coq7* and *Coq9* were knocked down. Inhibition of lipolysis by insulin was defective in all models and provision of CoQ_9_ improved insulin-regulated inhibition of lipolysis, albeit to different extents between models. These data are included in Figure 4, Figure 4—figure supplement 1–E, Figure 5 and Figure 5 and Figure 5 and Figure 5—figure supplement 1 and are discussed in the revised manuscript.

We measured circulating non-esterified free fatty acids (NEFAs) during an IP GTT in mice fed chow or HFHSD and treated with saline or 10 mg/kg CoQ_10_. These data are included in Figure 4 and Figure 4—figure supplement 1. Suppression of circulating NEFAs was slower in mice fed a HFHSD and administration of CoQ improved NEFA suppression during the GTT toward the kinetics observed in chow-fed mice. These data are discussed in the revised manuscript.

The Materials and methods section has been updated to include these methods.

Together these data strengthen the association between loss of mitochondrial CoQ and insulin resistance in adipocytes, showing that loss of CoQ impairs multiple insulin-regulated processes in adipocytes, including inhibition of lipolysis and stimulation of glucose transport.

2) Along the same lines, since the authors claim that CoQ supplement improves insulin sensitivity, it would be important to perform ITTs to support this.

These data are included in the revised manuscript.

Mice were fed a chow or HFHSD for 14 d with and without CoQ10 treatment. Mice were then fasted, anaesthetised and subjected to an insulin tolerance test with^[3^H]-2DOG tracer to allow determination of insulin-stimulated glucose uptake into specific tissues. These data are included in Figure 4. In chow-fed mice, CoQ had no effect on blood glucose during the ITT or tracer uptake into epididymal fat, inguinal fat or quadriceps. In contrast, CoQ improved insulin action in HFHSD-fed mice as evident by a greater lowering of blood glucose following insulin injection and increased tracer uptake into epididymal fat, inguinal fat and quadriceps. These data are discussed in the revised manuscript.

The Materials and methods section has been updated to include these methods.

These data strengthen our conclusions regarding a role for loss of CoQ in insulin resistance in adipose and muscle tissue in response to a HFHSD.

3) The improvement in insulin sensitivity upon catalase overexpression and complex II inhibition seems modest (Figure 6, Figure 7). To strengthen the causality between CoQ levels and insulin resistance it would be helpful to show whether TTFA/malonate treatment also improves insulin sensitivity in adipose explants derived from HFHSD-fed mice.

These data are included in the revised manuscript.

We have addressed this point in two ways: (1) strengthening the link between complex II activity in insulin resistance in our in vitro models and (2) applying this to adipose explants as suggested.

In our in vitro models (3T3-L1s treated with CI, Dex, TNF), we inhibited complex II with TTFA or malonate. These inhibitors lowered PRDX3 dimer/monomer ratio, and augmented insulin-stimulated HA-GLUT4 translocation only in insulin-resistant cells. These data are included in Figure 7 and Figure 7—figure supplement 1.

We also treated adipose explants with TTFA. TTFA had no effect on insulin-stimulated 2DOG uptake in explants from chow-fed mice, but improved 2DOG uptake in explants from HFHSD mice. These data are included in Figure 7.

These data strengthen our conclusions regarding a role for complex II activity-dependent oxidant production in adipocyte insulin resistance.